



# Dynamic performance of a passively self-adjusting floating wind farm layout to increase the annual energy production

Mohammad Youssef Mahfouz[1], Ericka Lozon[2], Matthew Hall[2], and Po Wen Cheng[1]

[1]Stuttgart Wind Energy at University of Stuttgart, Allmandring 5B, 70569 Stuttgart, Germany
[2]National Renewable Energy Laboratory (NREL), 15013 Denver W Pkwy, Golden, CO 80401, United States

**Correspondence:** Mohammad Youssef Mahfouz (mahfouz@ifb.uni-stuttgart.de)

**Abstract.** One of the main differences between floating offshore wind turbines (FOWTs) and fixed-bottom turbines is the angular and translational motions of FOWTs. When it comes to planning a floating wind farm (FWF), the translational motions introduce an additional layer of complexity to the FWF layout. The ability of a FOWT to relocate its position represents an opportunity to mitigate wake losses within an FWF. By passively relocating downwind turbines out of the wake generated by upwind turbines, we can reduce wake-induced energy losses and enhance overall energy production. The translational movements of FOWTs are governed by the mooring system attached to it. The way a FOWT relocates its position changes if the design of the mooring system attached to it changes. Additionally, the translational motion of a FOWT attached to a given mooring system is different for different wind directions. Hence, we can tailor a mooring system design for a FOWT to passively control its motions according to the wind direction. In this work, we present a new self-adjusting FWF layout design, and assess its performance using both static and dynamic methods. The results show that relocating the FOWTs in an FWF can increase the energy production by 3% using a steady-state wake model and 1.4% using a dynamic wake model at a wind speed of 10 m/s. Moreover, we compare the fatigue and ultimate loads of the mooring systems of the self-adjusting FWF layout design to the mooring systems in a current state-of-the-art FWF baseline design. The comparison shows that with smaller mooring system diameters, the self-adjusting FWF design has similar fatigue damage compared to the baseline design with bigger mooring system diameters. Finally, the ultimate loads on the mooring systems of the self-adjusting FWF design are lower than those on the mooring systems of the baseline design.

## 1 Introduction

To achieve our energy goals and harness more wind energy in sites with high wind resources, we cluster wind turbines into wind farms. However, this leads to energy losses inside the farm due to the generation of wakes. As the ambient wind field passes through a wind turbine, the turbine extracts energy out of the wind field, increasing its turbulence and decreasing its speed. When this lower speed and higher turbulence wind field reaches a downwind turbine, the turbine produces less energy and suffers from higher fatigue loads compared to the upwind one. The differences between the energy produced by the upwind and downwind turbines inside a wind farm are known as wake losses. For fixed-bottom onshore and offshore wind farms, the current state of the art is to optimize the wind farm layout to decrease the wake losses and increase the farm's annual energy





production (AEP) (Baker et al., 2019). Other wind farm layout optimization techniques focus on decreasing the cable length within the wind farm (Fleming et al., 2016) or using open multidisciplinary design, analysis, and optimization to decrease the wind farm's levelized cost of energy by optimizing the different components inside the wind farm's complex multidisciplinary system (Perez-Moreno et al., 2018). However, to the best of our knowledge, none of the current wind farm layout optimization techniques account for the ability of a floating offshore wind turbine (FOWT) to relocate its position in the horizontal plane.

For a FOWT, the mooring system is responsible for station-keeping, as it provides stiffness in surge, sway, and yaw degrees of freedom (DoFs). The mooring system allows the FOWTs to move within a constrained region called the watch circle. As the mooring system design changes, the stiffness of the FOWT's station-keeping system changes, and the shape of the watch circle changes. The ability of a FOWT to relocate its position is a new DoF that should be considered inside a floating wind farm (FWF) for two reasons. First, the translational motions of a FOWT have an effect on the FWF's AEP. Not considering these displacements increases the uncertainty in the FWF AEP calculations and hence increases the risk of investment (Bodini et al., 2020). Fleming et al. (2015) show that if a downwind turbine moves in the crosswind direction with a displacement of $20\%$ of its rotor diameter ($D$), the energy produced will change by $4\%$. The motion can either increase or decrease the FWF's AEP depending on the direction of the motion into the wake or out of the wake. The second reason is that the ability of FOWTs to relocate their positions can be used to increase the FWF energy production.

Several methods have been proposed to achieve beneficial crosswind motions in an FWF. Kheirabadi and Nagamune (2019) and Kheirabadi and Nagamune (2020) implemented a yaw and induction based turbine controller to reposition downwind turbines out of the wake. The idea was to use the FOWT controller to create a crosswind aerodynamic force component by yawing the FOWTs' nacelles to move the downwind turbines out of the wake. The results showed that relocating the FOWTs will increase the FWF's efficiency, and that the increase in efficiency is dependent on the mooring system designs because they govern the FOWTs' motions. Rodrigues et al. (2015) changed the positions of FOWTs inside an FWF using a pulley to change the length of the mooring lines according to the wind direction. The results also showed that relocating the FOWTs can increase the FWF's energy production. These previous studies aimed to actively control FOWT positions and showed the dependency of the FOWT motions on the mooring system design. In our previous work (Mahfouz and Cheng, 2023), we proposed a new method to passively relocate the FOWT out of the wake through mooring system design. However, we only used static tools to assess the method, and no dynamic verification was carried out. A summary of these three methods is given in Table 1.

In this paper, we build on our previous work (Mahfouz and Cheng, 2023), in which we presented a methodology to passively relocate the FOWTs in an FWF to increase the farm's AEP. The methodology customizes the mooring system design of each FOWT to increase the FWF's AEP by allowing larger platform offsets and reducing farm wake losses. Mahfouz and Cheng (2023) showed that we can passively relocate the turbines and quantified the gain in AEP we can achieve by passively relocating the FOWTs. However, until this point all our work was done using static tools focusing only on increasing the AEP gain without assessing the performance of the FWF layout design under dynamic wind and wave loading. In this paper, we extend our work to verify the results using dynamic models. Our goal is to assess the performance of our FWF layout design in both operational and extreme loading conditions. We will evaluate whether the larger displacements of the FOWTs increase the fatigue and extreme loads for the novel mooring system designs compared to a traditional, state-of-the-art, baseline mooring



**Table 1.** Different methods for relocating the FOWTs

| Method | Description | Relocates the FOWTs actively or passively? |
|---|---|---|
| Yaw and Induction Based Turbine Repositioning (YITuR) | Kheirabadi and Nagamune (2019) designed a yaw and induction based FOWT controller to relocate the FOWT in the crosswind direction. The idea is to create a thrust component in the crosswind direction that moves the FOWT. | Active |
| Movable FOWT Design | Rodrigues et al. (2015) created a new FWF layout design where the FOWTs change their position according to the wind direction. The FOWTs use a pulley attached to the floating platform to change the mooring line length and hence change their position. | Active |
| Passively relocating FOWTs using mooring system | In our previous work (Mahfouz and Cheng, 2023), we created a new FWF layout design method where the mooring system design is part of the FWF layout optimization. Each FOWT is attached to a customized mooring system that relocates it according to the wind direction. | Passive |

system. Moreover, we will verify the results of the steady-state wake model, which does not consider the FOWT's motions, against the dynamic wake model, which considers the FOWT's motions.

To accomplish this, we first present the baseline layout, the passively self-adjusting FWF layout design, and the energy gain we expect to achieve using static tools. Next, we present the stiffness and natural frequency values of each mooring system design within the FWF. Then, we compare the watch circles of the mooring system designs obtained using MoorPy with those obtained using OpenFAST. We compare the energy gain between steady-state and dynamic wake models at a constant wind speed of 10 m/s. We also calculate the fatigue damage on the mooring system designs and compare it to the fatigue damage of a baseline mooring system design. Finally, we assess the loads on the new mooring system designs under extreme wind and wave conditions to determine their ability to withstand ultimate loads.

## 2  Self-adjusting floating wind farm design

In this section we present the methodology to design the passively self-adjusting layout starting from a baseline wind farm layout. Additionally, we show the results of the static models for each step of the design process. The design process consists of four main steps as shown in Fig. 1. We discuss each step in the following sections. Throughout this work, we use the International Energy Agency Wind Technology Collaboration Programme (IEA Wind) 15 MW reference wind turbine (Gaertner





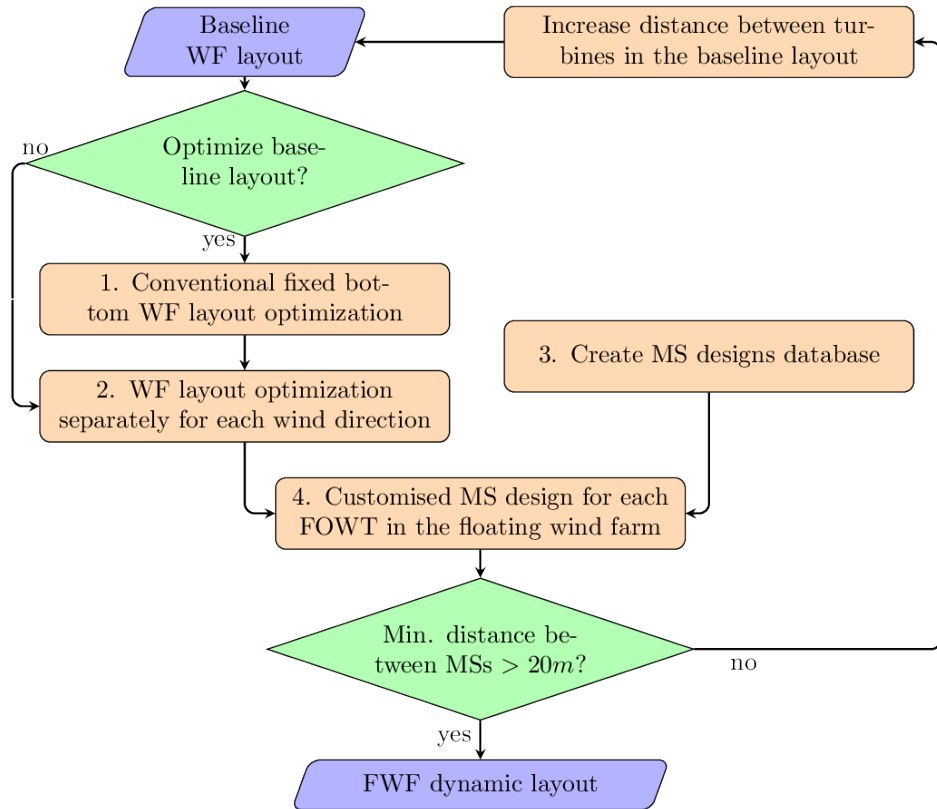

**Figure 1.** Workflow of designing a passively self-adjusting FWF layout

et al., 2020) coupled to the Activefloat semisubmersible floater as presented by Mahfouz et al. (2021). The wind rose intro-
duced within IEA Wind Task 37 (Baker et al., 2019) presented in Fig. 2 was used in this study. Throughout this work, the wind
speed was assumed to be constant in all wind directions and equal to 10 m/s. We chose the value of 10 m/s to be just below
the rated wind speed of 11 m/s of the FOWT reference model, as wakes have no effect on the wind farm's energy production
above the rated wind speed.

## 2.1 Conventional layout optimization

The first step in the novel FWF design methodology is to optimize the wind farm layout using the conventional state-of-the-
art methods assuming a fixed-bottom layout. This step can be skipped if the user provides a baseline layout that is already
optimized. We start with this step because our goal is to show the gain in a wind farm's AEP due to relocating the FOWTs.
Starting from a baseline layout far from the optimum will overestimate the effects of relocating the FOWTs showing higher
AEP gain. Instead, we compare the novel passive relocation technique to more traditional layout optimization methods with
limited floater motions. The baseline wind farm layout we started with in this paper is shown in Fig. 3. The layout has a
square shape and consists of nine wind turbines. We optimized the layout using the gradient-based optimizer SNOPT (Gill



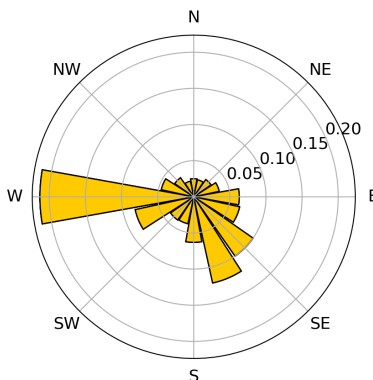

**Figure 2.** Wind rose from IEA Wind Task 37 with constant wind speed of 10 m/s

et al., 2005, 2008), the objective function was to maximize the wind energy production. The optimization process had two constraints. The first was to keep all turbines within the wind farm boundaries (i.e., within a square of side length $12D$). The mooring system designs are not included in this step; hence, when the mooring systems of all FOWTs in the FWF are designed, the anchors will not be within the FWF boundary. The second constraint was that the minimum distance between any two turbines in the wind farm cannot be less than $4D$.

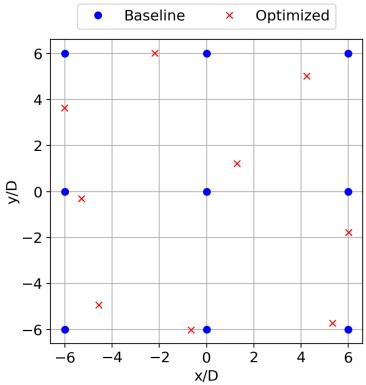

**Figure 3.** Baseline layout and optimized wind farm layout

The wakes were estimated during the optimization using FLORIS v3.1 (NREL, 2023). The wake model used was the Gaussian velocity model implemented within the tool based on the works of Bastankhah and Porté-Agel (2016); Niayifar and Porté-Agel (2016), and Gauss deflection model based on the works of Bastankhah and Porté-Agel (2016) and King et al. (2021). The turbulence model used was the Crespo-Hernandez model (Crespo and Hernández, 1996). The wake deflection in FLORIS only accounts for the horizontal wake deflection due to yawing the wind turbines. In this work, the yaw angle is always



defined to be zero for all wind turbines within the wind farm; hence, no horizontal wake deflection occurs for all calculations done using FLORIS. The optimized wind farm layout is shown in Fig. 3. We will refer to this layout as the optimized wind farm layout through the rest of the text. This layout will be used as a baseline to compare the energy gain after relocating the FOWTs in the FWF. The goal is that the customized mooring system will be coupled to the optimized wind farm layout and passively relocate the FOWTs to decrease wake losses.

### 2.2 Wind farm layout optimization per wind direction

The next step in the self-adjusting FWF layout design method is to find the FWF targeted layout. The targeted layout is the optimum wind farm layout we can achieve if we allow each turbine inside the wind farm to displace in the crosswind direction for every wind direction. To achieve the targeted layout we used the optimization algorithm SNOPT and the same wake model we used to get the optimized wind farm layout. We optimized the wind farm layout separately for each wind direction in the wind rose. The optimization objective was to increase the energy production in every wind direction. The optimization process had two constraints: First, the FOWTs were only allowed to displace in the crosswind direction. Second, the crosswind displacement could not be bigger than $0.5D$, which is a distance of $120$ m for the $15$ MW reference wind turbine.

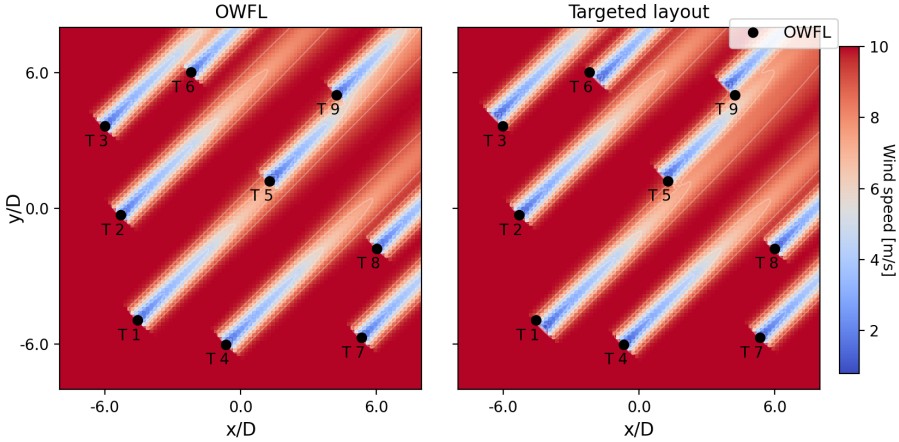

**Figure 4.** Wind farm layouts with wakes modeled by FLORIS: (left) the optimized wind farm layout and (right) the targeted layout. The black dots represent the positions of the turbines from the optimized layout.

Figure 4 shows results of the targeted layout for only one wind direction; this process was repeated for each wind direction. We can see that the turbine positions are shifted in the crosswind direction to decrease wake interactions. The energy gain of the targeted layout compared to the optimized layout for each wind direction is shown in Fig. 11 and was equal to $6.1\%$ energy gain at a constant wind speed of $10$ m/s. The energy gain was calculated using equation 1:

$$E_{gain} = \sum_{i=1}^{N} \frac{E_i - E_{0,i}}{E_{0,total}}, \tag{1}$$



where $E_i$ is the energy of the new layout (targeted layout) at a single wind direction $i$, and $E_{0,i}$ is the energy of the optimized layout at a single wind direction $i$, $E_{0,total}$ is the total energy of the optimized layout for all wind directions $N$, which is equal to the number of wind directions in the wind rose in Fig. 2. The targeted layout is not realistic and is impossible to achieve, as we cannot freely move the FOWTs to the displacements in all wind directions. Realistically, our goal is to get the FOWTs to displace close to the targeted layout for as many wind directions as possible. However, the FOWT displacements are governed

by the mooring system design, which cannot achieve the targeted displacement for all wind directions.

## 2.3 Mooring system database

The third step in the self-adjusting FWF layout design method as show in Figure 1 is to create a mooring system design database. The goal of the database is to contain several mooring system designs with different design parameters and save the watch circle and the displacements of each mooring system design for all wind directions. The database will be used in the next

step to find a customized mooring system design for each FOWT in the FWF. To create the mooring system design database, we created a full factorial design matrix iterating over several design parameters while keeping other parameters fixed. The fixed parameters were the bathymetry ($depth$), which was set at a value of 200 m; the number of mooring lines in each mooring system design, which was set to three lines; and the line type—all lines were assumed to be studless chains made of R4S steel.

The permutable design parameters used to create the full factorial mooring system design matrix were the following:

– Mooring line diameters: three nominal diameter values of 0.09 m, 0.12 m, and 0.15 m were used.

– Mooring line headings: 72 combinations of mooring system line headings were used covering all possible heading combinations, with an increment of $10°$, that a three-line mooring system design can have. The minimum angle between any two mooring lines was $10°$.

– Anchoring radius: two anchoring radius values of $2.5D$ and $3.5D$ were used. The anchoring radius is the horizontal

distance from the tower's center to the anchor.

– Mooring line length: three values of the mooring line length as a function of the anchoring radius were used, as shown in equation 2. In equation 2, $R$ is the horizontal distance between the fairlead position and the anchor position, $L_{max}$ is the maximum length a mooring line can have, and $L_{min}$ is the minimum length a chain mooring line can have. The three values of $\beta$ used for the mooring system design matrix were 0.5, 0.7, and 0.9. The line profiles of the three values of $\beta$

at the two anchoring radii are shown in Fig. 5.

$$L_{min} = \sqrt{depth^2 + R^2} \tag{2a}$$

$$L_{max} = depth + R \tag{2b}$$

$$L = L_{min} + \beta(L_{max} - L_{min}) \qquad 0 \leq \beta \leq 1 \tag{2c}$$

After defining these permutable design parameters, we ended up with a full factorial design matrix containing $419,904$

possible mooring system designs. We used MoorPy (Hall et al., 2021) to calculate the watch circle of each mooring system

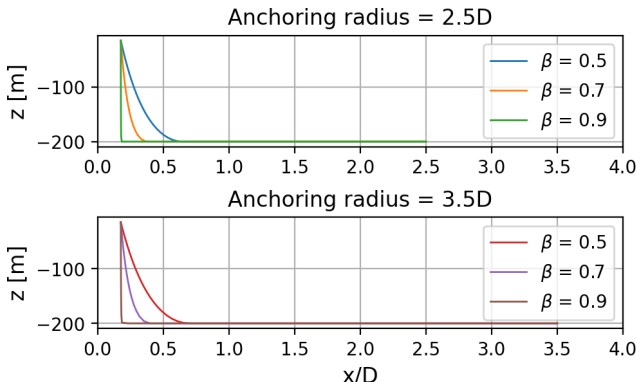

**Figure 5.** The line lengths and anchoring radii used while creating the database.

design. MoorPy is a quasi-static Python-based tool for mooring system analysis and design. In MoorPy a FOWT is modeled as a point mass with 6 DoFs and its hydrostatic properties. Additionally, an external force vector was applied to the FOWT at water level. This external force represented the aerodynamic thrust force vector at a wind speed of 10 m/s. We used coordinate transformation to transform the aerodynamic force from the FOWT's hub to the water level. This meant that the aerodynamic

forces applied at the water level were functions of the roll, pitch, and yaw motions of the platform. Also, the roll, pitch, and yaw motions affect the aerodynamic force vector at the hub coordinate system, as these motions change the aerodynamic angle of attack and alter the aerodynamic force vector. To account for these effects we assumed that the aerodynamic forces change linearly with the changes in roll, pitch, and yaw DoFs. We used OpenFAST (Jonkman, 2009) to quantify the changes in the aerodynamic forces per one degree change in roll, pitch, and yaw motions for the 15 MW rotor. In reality, the effects of

the FOWT's motions on the rotor aerodynamics are not linear, but this simplification was enough, and the results in MoorPy showed good agreement with OpenFAST as presented in Section 3.3.

In MoorPy, we iterated through all wind directions with an increment of 5° and found the steady-state position of each mooring system design for each wind direction. To account for the effect of FOWT motion on the aerodynamic force vector, we updated the aerodynamic force vector whenever the FOWT displaced until the solution converged. It is necessary to include

the effects of FOWT rotations on the aerodynamic forces and in the coordinate transformation so the watch circles produced in MoorPy will better match the results from fully coupled aero-elasto-servo-hydro simulations. While creating the mooring system database the wave loads were neglected, as their effect on the steady-state displacement of the FOWTs is negligible compared to the aerodynamic loading.

After calculating the mooring system watch circles and displacements for each design in the design matrix, we checked that

the designs passed the following constraints:

– The mooring system yaw angles are between −5° and 5°. We chose this constraint because a higher value will lead to high energy losses in the FWF due to yaw misalignment. We are not implementing any yaw control routines on a turbine





level or a farm level within this paper; hence, we decided to keep the mean yaw value within these limits to avoid yaw misalignment.

– The mooring system cannot displace more than $1D$ in the wind direction.

   – No vertical loads on the mooring system anchors. This means that all lines have catenary shapes with a section laying on the seabed for all wind directions.

After applying these constraints, we ended up with $440$ mooring system designs that we used to choose the customized mooring system for each FOWT in the FWF. We presented the effect each mooring system design parameter has on the shape of the

watch circle and the mooring system stiffness in our work in Mahfouz et al. (2022).

### 2.4  Customized mooring system designs

After creating the mooring system database, we used the targeted layout design to choose mooring system with watch circles closest to the target displacements. This process was explained in detail in our work in Mahfouz and Cheng (2023). We used brute force optimization to find the mooring system design for each FOWT to increase the overall energy production of the

FWF. Afterward, we checked that the minimum distance between any two mooring lines within the FWF was larger than $20$ m as required in the API (2005) standard. If the distance between two mooring lines was less than $20$ m, we increased the minimum acceptable distance between any two turbines while designing the optimized wind farm layout in Section 2.1. In this work, we started with the constraint of minimum distance equal to $2D$, and we iterated till the required minimum distance between any two lines was larger than $20$ m. Therefore, we ended up with a constraint of minimum distance equal to $4D$ for the

optimized layout design as introduced in Section 2.1. The design parameters of the nine customized mooring systems attached to each FOWT in the FWF can be found in Table A1 in the appendix.

  The positions of the final self-adjusting FWF layout relative to the optimized wind farm layout for one direction are shown in Fig. 6, which shows how the crosswind movement of the FOWTs in the layout decreases wake interactions inside the FWF as Turbines $9$ and $6$ move out of the wake of Turbines $5$ and $3$, respectively. The energy gain of final FWF layout design compared

to the optimized layout at each wind direction is shown in Fig. 11. As expected, the total energy gain of the final FWF is only $3.1\%$ compared to $6.1\%$ of the targeted layout, as the mooring system designs cannot achieve all crosswind displacements like the targeted wind farm layout design. We can see in Fig. 6 that turbines are displaced in the crosswind direction, and in-wind displacements are also visible, because the FOWTs are coupled to the mooring system designs. An overall view of the floating optimized wind farm layout with the mooring system attached to each FOWT is shown in Fig. 7.

## 3  Design performance and verification

In this section we verify the initial results that we obtained using static wind loading and a steady-state wake model against dynamic wind and wave loading and a dynamic wake model. We use the mid-fidelity, state-of-the-art aero-servo-elasto-hydro tool OpenFAST v3.4.1 to model the FOWT and assess its behavior in operational and extreme conditions. Additionally, we use



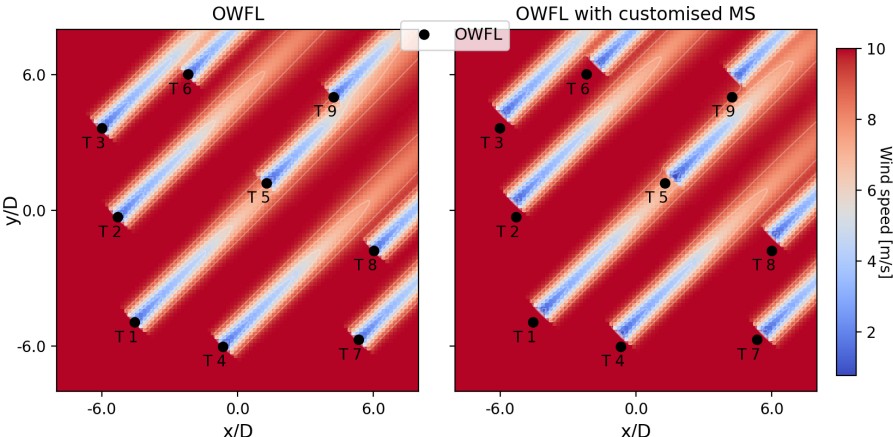

**Figure 6.** Wind farm layout and wakes: (left) the optimized layout and (right) the final layout. The black dots represent the positions of the turbines in the optimized layout.

FAST.Farm v3.4.1 (Jonkman and Shaler, 2020) to model the FWF and assess how much energy gain relocating the FOWTs
will achieve using the dynamic wake meandering (DWM) model.

## 3.1 Introducing baseline floating wind farm design

Before starting to assess and verify the novel FWF layout design performance, we introduce a baseline mooring system design
that we use as a benchmark to compare the customized mooring system designs against. A baseline is needed to reflect the
difference between the current state-of-the-art mooring system designs with small translational displacements (a few meters)
and the customized mooring system designs that relocate the FOWTs more than 100 m.

The baseline mooring system design process assumed a fixed anchor spacing of 600 m, a water depth of 200 m, and three
mooring lines at evenly distributed heading angles. The maximum platform offset was constrained to be 12% of water depth,
about 25 m. The length and diameter of the chain were varied to meet constraints on extreme tensions and fatigue damage,
with safety factors of 2 and 3, respectively. The fatigue damage was checked with a simplified fatigue analysis that consisted
of three load cases, with wind speeds near the rated wind speed and corresponding waves. The load cases assumed aligned
wind and waves, and uniform distribution of wind-wave headings. The API (API, 2005) fatigue analysis process was followed,
using the T-N curve for chain. Fatigue was the driving constraint in the design of the baseline mooring system, resulting in a
chain diameter of 0.19 m. The properties of the baseline mooring design are shown in Table 2.

**Table 2.** Parameters of baseline mooring design

| # of Lines | Depth | Anchoring radius | Fairlead radius | Fairlead depth | Line type | Line length | Line diameter |
|---|---|---|---|---|---|---|---|
| | (m) | (m) | (m) | (m) | | (m) | (m) |
| 3 | 200 | 600 | 42.5 | 15 | Chain | 623.3 | 0.1902 |





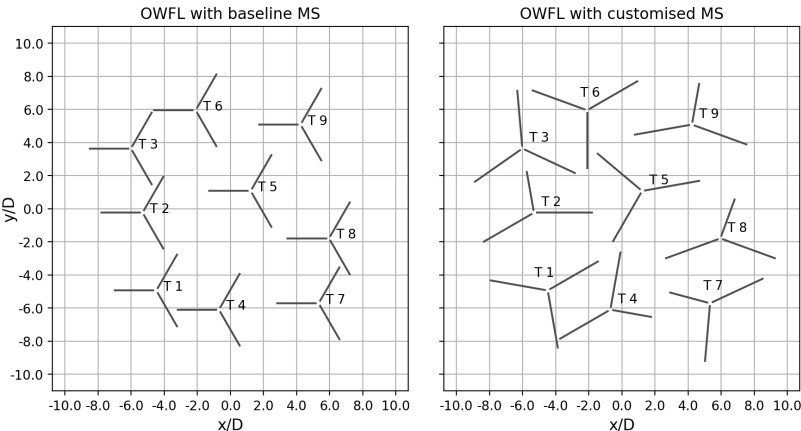

**Figure 7.** The optimized wind farm layout coupled to (left) the baseline mooring system and (right) the customized mooring system.

The optimized wind farm layout coupled to the baseline mooring system and the customized mooring system is shown in Fig.
7. Both layouts are shown in the absence of wind and wave loading. Mahfouz (2023) presents an animation video illustrating
the passive displacements of the FOWTs according to the wind direction. The video shows a top view of the wind farm layout
and the mooring systems and how the FOWTs move passively as the wind direction changes.

### 3.2 Customized mooring system stiffness

We compared the natural frequency of FOWTs when coupled to the customized mooring system designs versus the baseline
design. We used MoorPy to calculate the stiffness of the mooring system at each wind speed and wind direction to cover the
FOWT operation range between cut-in and cut-out wind speeds. Then, we used the results to calculate the natural frequencies
of the system over the watch circles of the mooring systems. The results for the baseline mooring system and the customized
mooring system design of the first turbine (T1) in the FWF are shown in Figs. 8, and 9, respectively. The figures show the value
of the natural frequencies in the $x$-axis and $y$-axis directions. Each dotted line in the plots represents the watch circle at one
wind speed; the most inner line is the watch circle at a wind speed of 3 m/s, and the outermost line is the watch circle at rated
wind speed of 11 m/s. The color of the plots represents the value of the natural frequency over the different positions of the
watch circle.

In Fig. 8, the natural frequency of the baseline design does not change as the wind speed and wind direction changes. This
means that for all wind excitations and for all positions inside the watch circle, the natural frequency is constant. This is because
the stiffness of the baseline design is linear and almost constant for all wind speeds and wind directions. On the other hand, Fig.
9 shows that the natural frequencies of the customized mooring systems are rather spread; the frequencies are low at low wind
speeds and increase to be within the same frequency range of the baseline mooring system design as the wind speed increases.
This is because the stiffness of the customized mooring system design is nonlinear and changes with the wind speeds and the

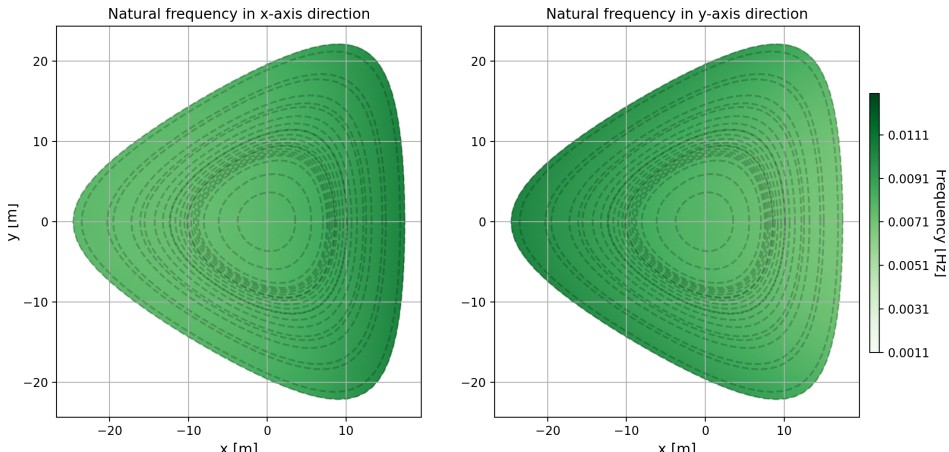

**Figure 8.** The natural frequency in $x$ and $y$ directions for the FOWT when coupled to the baseline mooring system. The dotted lines represent the watch circles at wind speeds ranging from 3 m/s to 11 m/s.

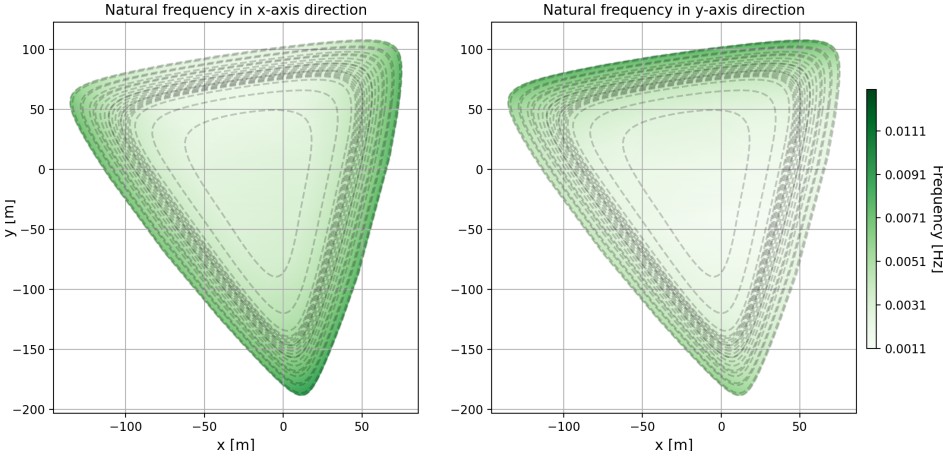

**Figure 9.** The natural frequency in $x$ and $y$ directions for the FOWT when coupled to the customized mooring system of Turbine 1. The dotted lines represent the watch circles at wind speeds ranging from 3 m/s to 11 m/s.

wind directions. In general, the frequencies of the customized mooring system are lower than the baseline design because their
stiffness values are lower to allow larger mean FOWT translational displacements.

### 3.3 Steady-state behavior

Before applying any dynamic loads on the FOWTs, we verified that the mean steady-state displacements calculated with
MoorPy were equal to the steady-state displacements in OpenFAST. This comparison was important before running FAST.Farm
simulations to quantify the energy gain of displacing the FOWTs in the FWF and comparing it to the results of the static model.





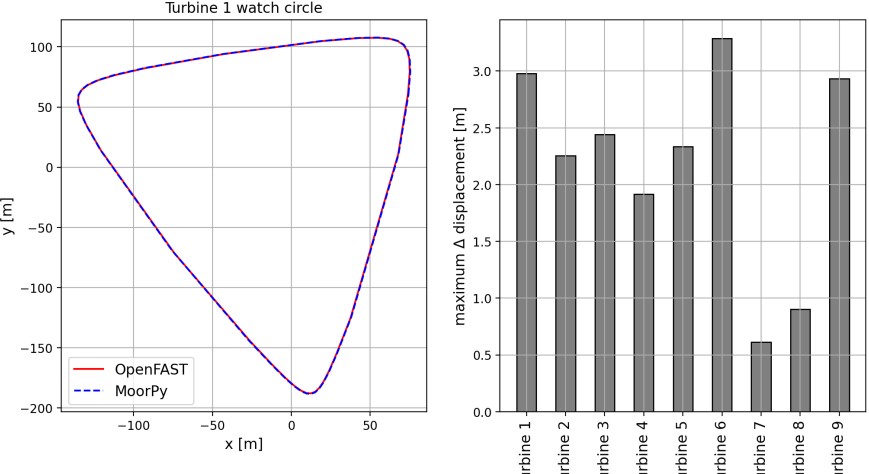

**Figure 10.** (left) The watch circle of Turbine 1 in both MoorPy and OpenFAST. (right) The maximum difference between the watch circle of each turbine in the FWF in MoorPy and OpenFAST.

If the mean displacements of the two models are different, it will lead to a different layout for every wind direction, and we cannot directly compare the energy gain of the two models. Hence, we applied a steady wind of 10 m/s at the rotor hub using OpenFAST; then, we compared the steady-state positions between the two models in Fig. 10. The figure shows the comparison of the watch circle of Turbine 1 in the FWF in both tools on the left side and the maximum difference between the watch circles produced by MoorPy and OpenFAST. The biggest difference in displacement between the two models is

around 3.3 m for Turbine 6. This difference is negligible compared to the 240 m rotor diameter of the 15 MW reference wind turbine. The reason for this difference is that OpenFAST models the full structure of the FOWT, captures the tower top deflections, and includes them while transforming the aerodynamic forces from the tower top coordinate system to the water level coordinate system. On the other hand, the MoorPy model is a simplified model and does not account for structural deflections; therefore, the forces applied at the water level are different between the two models. Moreover, we assumed in

MoorPy that the aerodynamic force vector at the hub height changes linearly with the FOWT's roll, pitch, and yaw angle, which is different than OpenFAST, which recalculates the aerodynamic forces using blade element momentum theory.

### 3.4 Energy production

In this section, our goal is to compare the FWF energy gain that we calculated using the steady-state Gaussian wake model in FLORIS with the results of the mid-fidelity DWM from FAST.Farm. FAST.Farm uses OpenFAST to account for the aero-elasto-

servo-hydro dynamics of each FOWT in the FWF and the DWM model to account for wake deficits, advection, deflection, meandering, and merging. We simulated the optimized wind farm layout twice at every wind direction of the wind rose in Fig. 2—first with all the FOWTs in the FWF coupled to the baseline mooring system design, and second with each FOWT coupled to its customized mooring system design. Our wind rose has 16 wind directions, resulting in 32 FAST.Farm simulations. All





simulations settings and the wind fields for both cases, with the baseline mooring system and with the customized mooring

systems, were identical for all wind directions.

The wind fields were generated using the Mann turbulence box (Mann, 1998). The Mann model parameters were defined following the IEC (2019) standard to achieve a turbulence intensity of $6\%$. The mean wind speed at the hub was equal to $10$ m/s. For all simulations, the resolution of the high-resolution grid in FAST.Farm in the $x$, $y$, and $z$ directions was equal to $5$ m, and the low-resolution grid's resolution was set to $35$ m. The low-resolution time step was set to $3.78$ s, while the high-resolution

time step was $0.42$ s, and the OpenFAST time step was set to $0.0378$ s. The significant wave height for all simulations was equal to $2$ m, and the wave period was equal to $6$ s. The second-order wave forces were not considered in any of these simulations. The simulation time was $4200$ s, and the first $600$ s were excluded to avoid transient effects. Additionally, the simulations started with the FOWT positions set according to the values obtained from MoorPy to avoid big transient movements. The polar formulation of the DWM in FAST.Farm was used with the default settings of the wake model. The default settings were

calibrated and verified by Doubrawa et al. (2018) and Jonkman et al. (2018) against large-eddy simulations for a fixed-bottom $5$ MW wind turbine. FAST.Farm wake parameters are not yet calibrated for FOWTs.

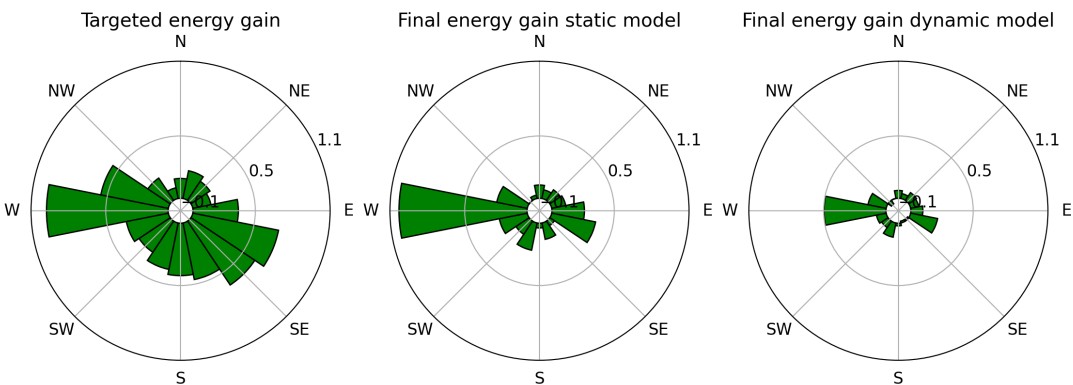

**Figure 11.** (left) The energy gain at each wind direction for the targeted wind farm layout, (center) the final wind farm layout using static wake models, and (right) the final wind farm layout using dynamic wake models.

After omitting the first $600$ s, the final $3600$ s of each simulation was processed. The average power produced by each turbine in both FWF layouts (with baseline mooring system and with the customized mooring system designs) was calculated. The energy gain of the optimized layout coupled to the customized mooring system was calculated relative to the optimized

layout when coupled to the baseline mooring system as in equation 1. The gain from FAST.Farm was equal to $1.4\%$, and the gain for every wind direction in the wind rose is shown on the right side in Fig. 11. The energy gain distribution per wind direction in the FAST.Farm model follows the same trend as the gain distribution for the MoorPy-FLORIS results. However, the gain values predicted by FAST.Farm are smaller than what the steady-state models predicted. The difference between the MoorPy-FLORIS static model and FAST.Farm model can be caused by two reasons. First, the FOWTs are not being displaced

by the customized mooring system designs as we predict in the design phase. However, this possibility is omitted because as



we presented in Section 3.3, the FOWTs relocate their positions in OpenFAST and MoorPy similarly and the difference is negligible. The second source of difference between the MoorPy-FLORIS model and the FAST.Farm model comes from the difference between the wake models implemented in FLORIS and FAST.Farm, as well as the difference in fidelity between MoorPy-FLORIS and FAST.Farm. First, FLORIS is a steady-state model that does not account for the FOWT's mean pitch

angle, which means that the rotor average wind speeds in FLORIS and OpenFAST are different. Moreover, Ramos-García et al. (2022b, a), Johlas (2021), and Nanos et al. (2022) showed that the positive pitch angle of FOWTs deflects the wake upward. This deflection is captured in FAST.Farm but not in FLORIS. Therefore, we analyzed the difference between the wind speed deficit in FLORIS and FAST.Farm to understand the difference between the models. Since the main energy gain happens when the wind is blowing from west to east (wind direction $= 0°$), we used this wind direction during this analysis.

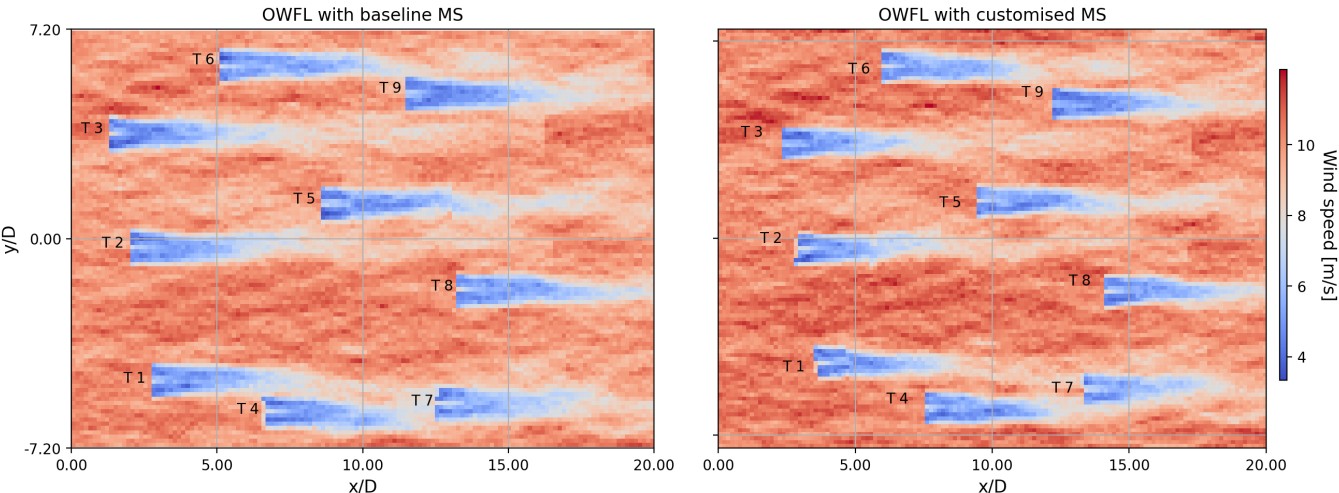

**Figure 12.** The wind flow field at the rotor hub at $4200$ s. (left) The non-relocating optimized wind farm layout coupled to the baseline mooring system design compared to (right) the self-relocating optimized layout at wind direction $= 0°$.

The wind flow field at the rotor hub from FAST.Farm is presented in Fig. 12—the flow field of the optimized wind farm layout with the baseline mooring system on the left and the self-relocating layout on the right. Figure 13 shows the rotor average wind speed in FLORIS and in FAST.Farm for a wind direction of $0°$. On the left, all FOWTs are coupled to the baseline mooring system with limited movement in the crosswind direction; in the middle, each FOWT is attached to its customized mooring system design, allowing crosswind movements; and on the right is the difference in rotor average wind speed of each turbine.

Turbines 7 and 9 are relocated out of the wake, and their rotor average wind speeds are higher, as shown in Fig. 13. From Fig. 13 we can conclude the following:

– The rotor average wind speed of a FOWT is always slightly lower in FAST.Farm when compared to FLORIS, because FAST.Farm considers the pitch, roll, and yaw motions while FLORIS cannot capture them. For an inflow wind field with mean wind speed of $10$ m/s, the average rotor wind speed of FOWTs in FAST.Farm is $9.5$ m/s, instead of $10.0$ m/s in

FLORIS.





– In FLORIS, Turbine 4 is partially in the wake of Turbine 1 in the baseline floating optimized layout, while in FAST.Farm this is not the case. This can be caused by wake meandering, which is considered in the FAST.Farm model and not in FLORIS, so the wake is deflected and Turbine 4 is no longer in the wake of Turbine 1. It can also be caused by horizontal wake deflection, as the platform yaw angle of Turbine 1 will deflect the wake away from Turbine 4. However, this reason is omitted, as the FOWT yaw angle of Turbine 1 is almost zero.

– Looking at the change in rotor average wind speeds between the optimized layout when attached to the customized mooring system designs and the baseline mooring system design, we can see that there are small differences in the rotor average wind speeds for the turbines in the free flow wind fields (such as Turbines 1, 2, 3, and 6). These differences come from the difference in the mean pitch angle of the FOWTs affected by the mooring system design overhang. The difference in the mooring system designs has a slight effect on the floating platform's pitch stiffness and affects the steady-state pitch angles of FOWTs.

– The wake losses in FAST.Farm are less than in FLORIS. This is clear from the rotor average wind speed of Turbine 7 in both FAST.Farm and FLORIS. In Fig. 13, for the optimized layout with the baseline mooring system, the average rotor wind speed of Turbine 7 in the FLORIS model is lower than the FAST.Farm model, which indicates that the wind speed deficit is higher in FLORIS and hence there are higher losses. After the optimized layout is modelled with the customized mooring system designs and the turbines are relocated, the wind speed increases in both models, proving that relocating the layout decreases the wake losses for Turbine 7. The value of the rotor average wind speed of Turbine 7 after the FOWTs are relocated is slightly higher in the FAST.Farm model. However, the total increase in FLORIS is higher, and hence the energy gain of relocating Turbine 7 is also higher. This was seen for all the turbines in the wake for all wind directions.

These four differences between the FAST.Farm and the FLORIS models lead to the difference in the energy gain of relocating the wind turbines. The first three differences are expected, as they result from the lack of turbine structural modelling in FLORIS or the simplicity of the steady wake model compared to the DWM. However, the big difference in velocity deficit of the two models requires further analysis. To understand why FLORIS estimates higher wake losses than FAST.Farm and if this difference is because of the effect of FOWTs on wake recovery and wake deflection, we decided to simulate the optimized wind farm layout as a fixed-bottom wind farm and compare it to the optimized layout when coupled to the baseline mooring system design. The results of the average rotor wind speed of the fixed-bottom optimized layout in FAST.Farm are shown in Fig. 14.

Figure 14 shows that the FAST.Farm rotor average wind speeds of the turbines in ambient wind (Turbines 1, 2, and 3) for the fixed-bottom optimized layout are higher than the floating optimized layout but still lower compared to FLORIS. This difference comes from the rotor uptilt angle of the turbines. This uptilt angle is only accounted for in FLORIS in the power coefficients look-up table, and it does not show in the rotor average wind speeds. This means that for a the fixed-bottom turbine, the mean power produced in the ambient wind field in FLORIS and FAST.Farm is equal. Additionally, Fig. 14 shows that for the fixed-bottom optimized layout, the velocity deficit for Turbine 7 in FAST.Farm is equal to FLORIS and is higher than the





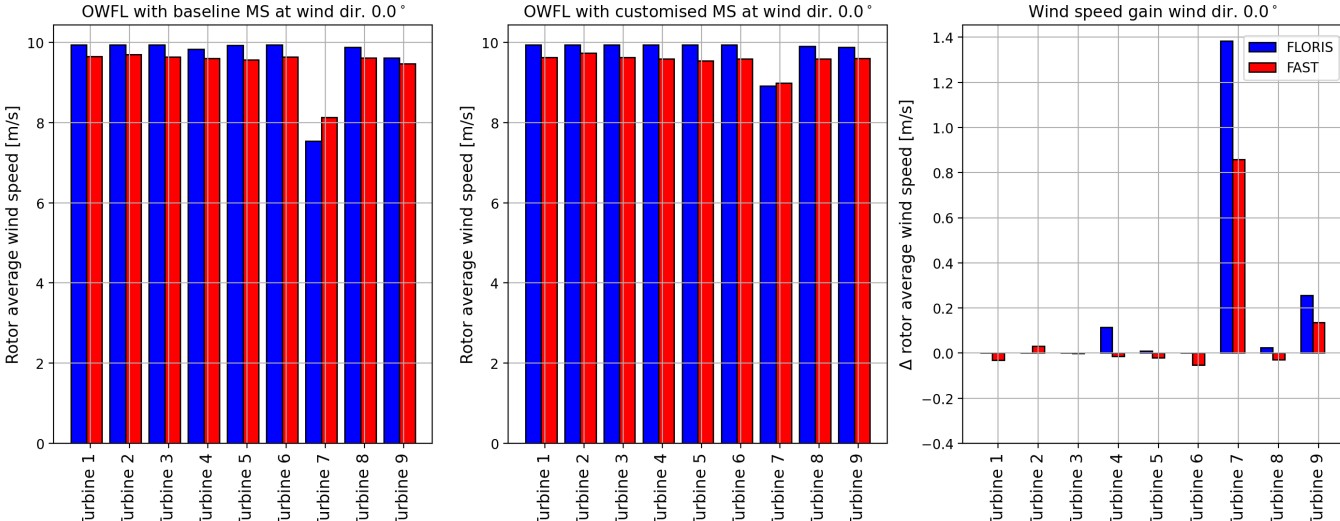

**Figure 13.** The rotor average wind speed in FLORIS and FAST.Farm at wind direction = $0°$. (left) The rotor average wind speed of the optimized wind farm layout coupled to the baseline mooring system. (center) The rotor average wind speed of the optimized layout coupled to the customized mooring system design. (right) The difference between the average rotor wind speed of the two configurations.

335 deficit of the floating optimized layout with the baseline mooring system design. This proves that the difference in velocity deficit is due to the motions of FOWTs and their effect on the wake rather than to the different fidelity of the steady wake model in FLORIS and the DWM in FAST.Farm.

 Figure 15 shows that the wake center of Turbine 4 is deflected upward when it is floating compared to when it is fixed. After a distance of $6D$, the wake center of Turbine 4 is at a height of $193$ m for a fixed-bottom turbine and a height of $234$

340 m for a FOWT. This means that the FOWT's wake is deflected $41$ m more compared to that of a fixed-bottom turbine, which is equivalent to $17\%D$. This agrees with the results found in Ramos-García et al. (2023), which indicated that FAST.Farm results showed larger vertical wake deflection when compared to the higher fidelity wake model presented by Ramos-García et al. (2022b). Moreover, Fig. 15 shows the horizontal wind field of the wind farm at the hub height, which shows that the wake recovery is faster inside an FWF compared to a fixed-bottom wind farm. This is clear when looking at the wakes behind

345 Turbines 2 and 3. The main reason for the faster wake recovery at the hub height is the vertical deflection of the wakes. Another reason is the motions of the FOWTs, which increase the turbulence of the flow field and hence increase the wake mixing and accelerate the wake recovery. In general, further work is needed to verify and calibrate the mid-fidelity DWM model in FAST.Farm with higher fidelity models to capture the effects of FOWTs on the wake with more certainty.

### 3.5 Performance in normal operation

350 To check if relocating the FOWTs increases the fatigue loads of the customized mooring system designs compared to the baseline mooring system, we calculated the damage equivalent loads (DELs) and the $25$-year fatigue damage at rated wind

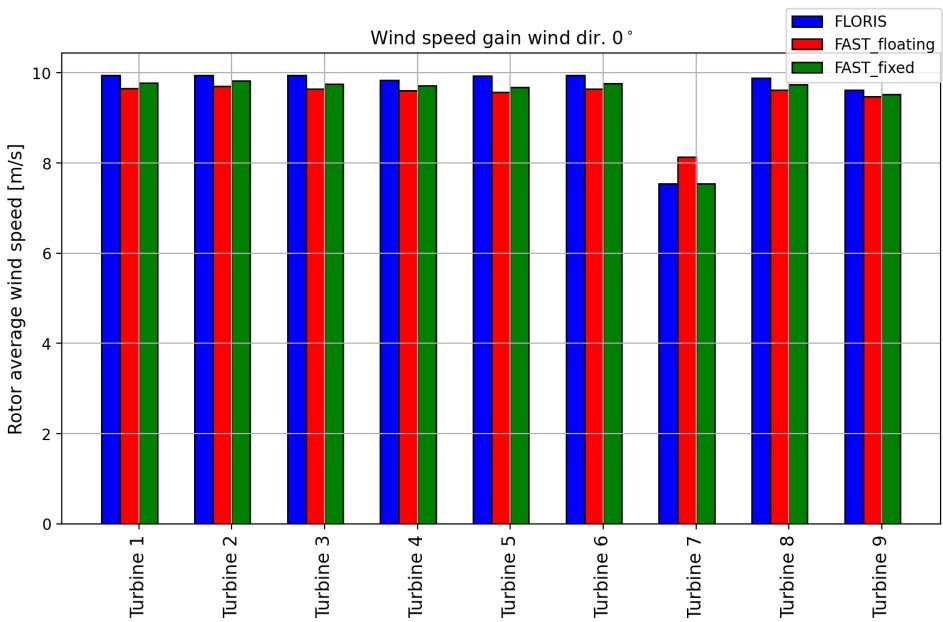

**Figure 14.** The rotor average wind speed of the optimized wind farm layout as fixed bottom and coupled to the baseline mooring system in FAST.Farm and FLORIS

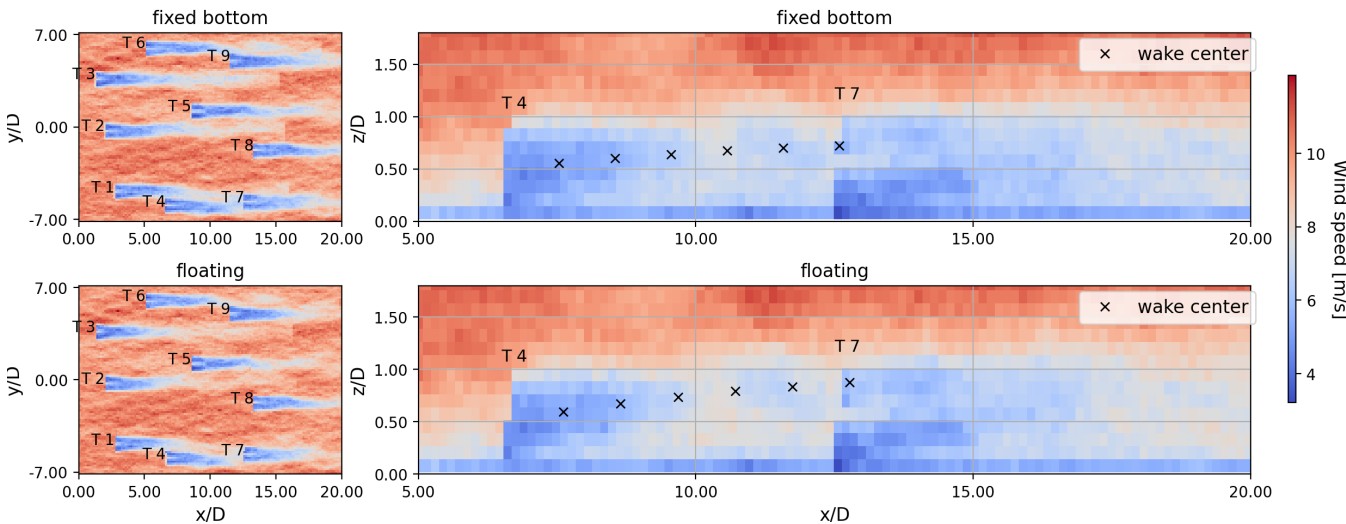

**Figure 15.** FAST.Farm results of (top) the optimized wind farm layout as fixed bottom and (bottom) the optimized wind farm layout coupled with the baseline. The left side represents the horizontal plane at hub height; the right side is the vertical plane passing through the hub of Turbine 7.

speed of 11 m/s. OpenFAST simulations of 4200 s were done for each of the nine turbines within the FWF. The wind field was generated using the Mann turbulence generator, and its parameters were defined following the IEC (2019) standard to achieve




a turbulence intensity of 6%. The fatigue loads of the mooring lines were assessed at all wind directions with a step of 10°

leading to 36 wind directions. The significant wave height for all simulations was set to 2 m, and the wave period was set to 6 s. The wind and wave headings were aligned for all wind directions. The second-order wave loads were considered for all directions, and one wave seed and wind seed were used for each wind direction. We chose to do the fatigue analysis at the rated wind speed, as from our experience, this is were the fatigue loads are the highest. In this work, we focused on estimating the mooring line fatigue and did not check the effect of relocating the turbines on the blades and tower fatigue loads. Moreover,

we did not check the wake effect on the mooring system fatigue loads, as it is out of the scope of this work.

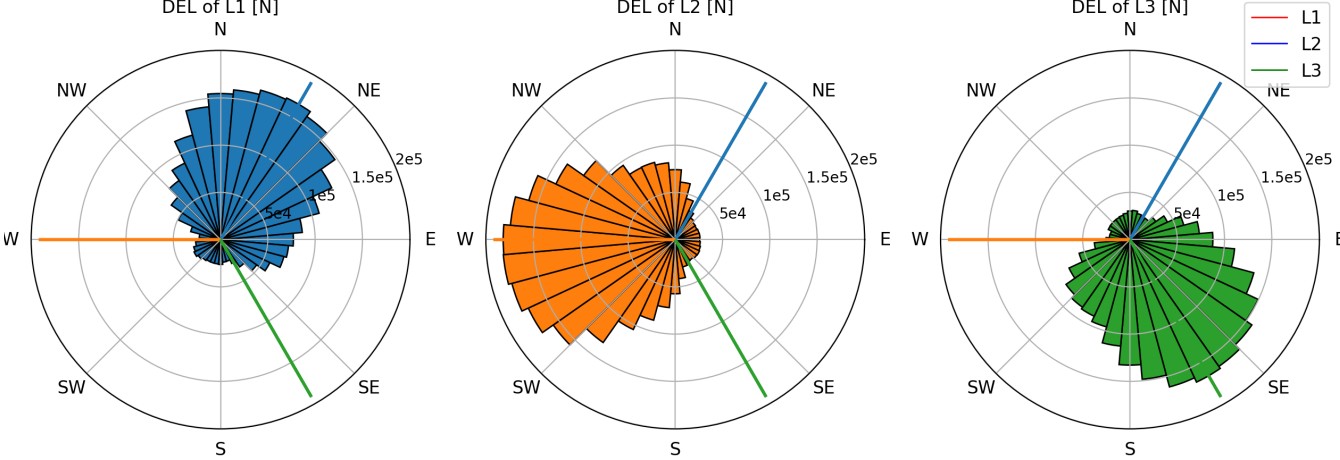

**Figure 16.** 1 Hz DEL of the baseline mooring system design at each wind direction

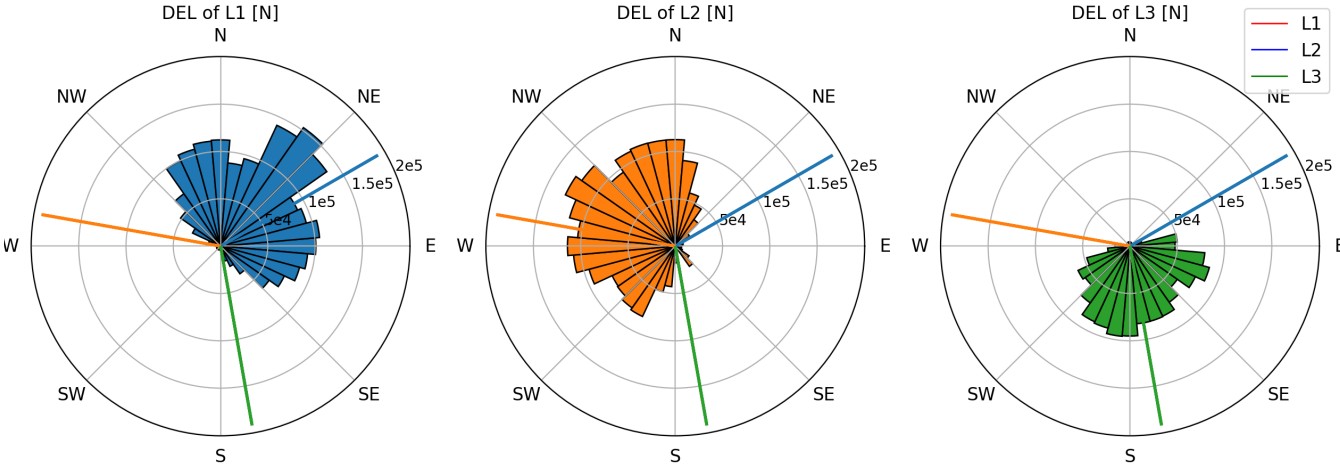

**Figure 17.** 1 Hz DEL of the customized mooring system design of Turbine 1 in the FWF at each wind direction





$$DEL = (\sum_{i=1}^{j} \frac{N_i}{T_{sim}} S_i^m)^{1/m} \tag{3}$$

The 1 Hz DELs were calculated using equation 3, with a Wohler exponent $m = 3$, and the $T_{sim}$ equal to 3600 because the first 600 s from each simulation were omitted to decrease transients effects. The rainflow counting method was used to count the load cycles for the mooring line fairlead tensions. The DELs of the three mooring lines of the baseline mooring system for

each wind direction are shown in Fig. 16. The maximum DEL of each of the three lines is when the wind direction is aligned with the line, as this line will be carrying the highest tension. For example, when the wind direction is from east to west, Line 2 of the baseline mooring system in Fig. 16 will have the highest DEL value. Moreover, since the three lines in the baseline mooring system are identical and mooring system is symmetrical, the DELs of the lines are also symmetrical. In Fig. 17 the DELs of the lines of the customized mooring system attached to Turbine 1 can be seen. Similar to the baseline mooring system,

the DELs are highest when the wind is aligned with a mooring line. However, since each mooring system is asymmetric, and the mooring lines are not identical, the values of the DELs of each line differ. In a wind farm design, we can benefit from the fact that the highest DELs happen when one of the lines is aligned with the wind direction to decrease fatigue loads. This can be achieved by avoiding having any of the mooring lines aligned with the most probable wind directions.

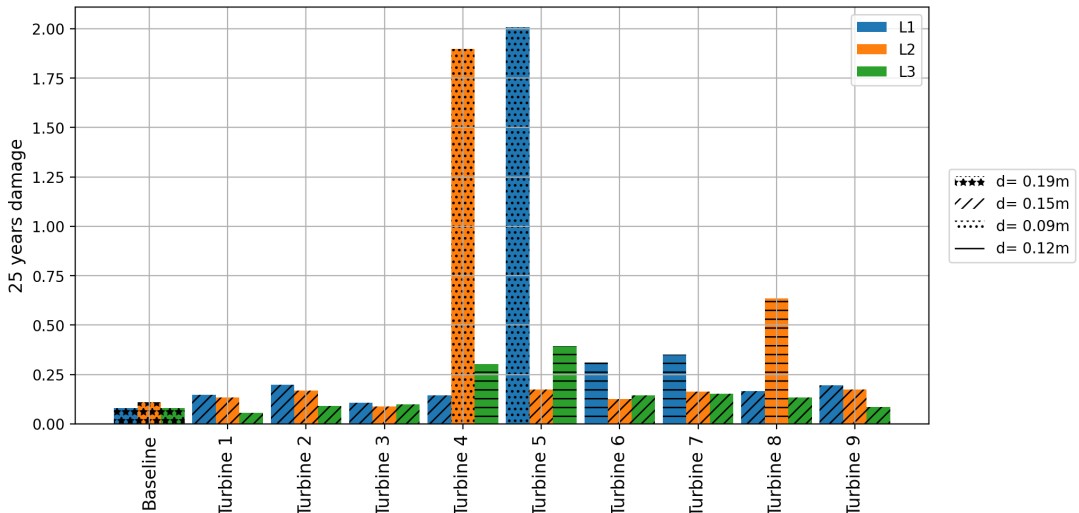

**Figure 18.** 25-year damage for all mooring system designs

$$N = k(\frac{T}{MBL})^{1/m} \tag{4}$$

$$MBL = 30.2D^2(44 - 80d) \qquad [MN] \tag{5}$$





We calculated the 25-year damage using the T-N curve for studless chain mooring lines as defined within the API (2005) standard in equation 4, with slope $m = 3$ and intercept $k = 316$. $T$ is the fairlead tension of the mooring lines, and $N$ represents the cycles to failures. The minimum breaking load (MBL) is a function of the mooring line diameter $d$ as in equation 5. For the damage calculations, we assumed that for the 25-year time, the wind speed is always at 11 m/s, and no other wind speeds where considered. This is enough for the purpose of the paper comparing the customized mooring system fatigue to the baseline mooring system fatigue, but it does not show whether or not the mooring system designs survive a 25-year lifetime. As each mooring system has different headings and hence a different DEL at each wind direction, we decided to remove the effect of wind direction and line headings on the damage. We assumed equal wind distribution over all 36 wind directions considered. The 25-year damage results are presented in Fig. 18.

The baseline mooring system fatigue damage is generally the lowest with the largest line diameter of 0.19 m. The customized mooring system 25-year fatigue damage is overall comparable, except for a few lines. The mooring lines with fatigue damage greater than 1.75 have a line diameter of only 0.09 m. The remainder of lines with fatigue damage greater than 0.25 have a line diameter of 0.12 m. All of the customized mooring lines with a diameter of 0.150 m show fatigue damage of less than 0.25, which is comparable to the baseline design. This is notable because the customized mooring system designs achieve similar fatigue damage with a significantly smaller line diameter.

The fatigue performance of the mooring system relates to the mooring system stiffness. The stiffness at a given position is equal to the slope of the force-displacement response of the mooring system at that offset. The FOWTs are offset to a mean displacement under the thrust force of the wind turbine. Then, waves act on the mooring system causing oscillations in position and tension about the mean. The stiffness, or the slope of the tension-displacement response, is representative of the amplitude of wave-frequency tension fluctuations about the mean. The customized mooring system designs have much lower stiffness overall compared to the baseline, resulting in lower tension amplitudes and resulting fatigue damage.

### 3.6 Performance in extreme conditions

The ability of the customized mooring system designs to withstand extreme loads was also checked. We applied extreme 50-year wind and wave conditions with a wind speed of 28.35 m/s, a significant wave height of 6 m, and a wave period of 11 s. These extreme values were chosen because they were the values used to design the Activefloat platform used within this work as introduced by Mahfouz et al. (2021). The extreme wind field was produced using the Mann turbulence generator with turbulence intensity of 11% as recommended by the IEC (2019) standard for extreme wind conditions. OpenFAST was used to simulate each of the customized mooring system designs and the baseline mooring system design for 36 wind directions, wind and waves were aligned for all directions, and second-order wave loads were considered. The simulation time was set to 4200 s, and the first 600 s were omitted to avoid transient effects.

The maximum tension of each line for all the 36 wind directions is shown in Fig. 19. The maximum tensions of the customized mooring system design are always lower than the tension of the baseline mooring system. Moreover, the maximum tensions were always lower than the maximum allowable tension for all mooring lines. The maximum allowable tension was calculated by dividing the MBL, as defined in equation 5, by a safety factor of 2. This safety factor is consistent with the safety





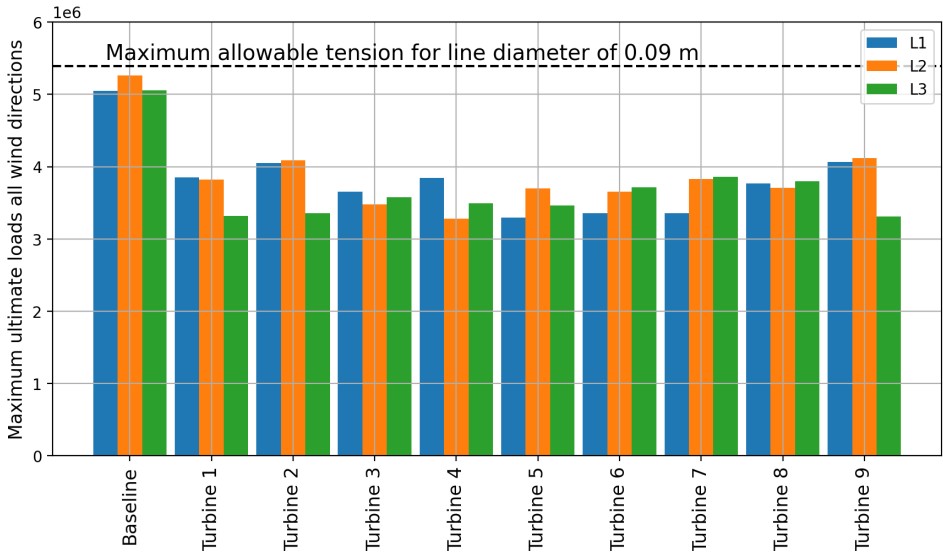

**Figure 19.** Maximum tension of each mooring system design for all wind directions under extreme wind and wave loads

factor defined by the API (2005) standard. The maximum allowable tension shown in Fig. 19 stands for the value for a mooring line of 0.09 m diameter, as this is the smallest diameter in the customized mooring system designs. Therefore, we can conclude from Fig. 19 that all mooring system designs can withstand the extreme loads.

## 4   Conclusions

This paper verifies an innovative method for FWF layout design and optimization using mid-fidelity dynamic models. The novel
method aims to passively relocate downwind FOWTs out of the wakes of upwind ones in an FWF to decrease wake losses. The relocation of the FOWTs depends on the mooring system design to govern the FOWTs' movements according to the wind direction. The verification process has two key objectives: First, verify the energy gain predicted by the Gaussian steady-state wake model implemented within FLORIS against the DWM model within FAST.Farm. Second, verify that the novel mooring system designs can withstand the fatigue and extreme loads in operational and extreme conditions when compared to
a state-of-the-art mooring system design.

The findings of the paper can be summarized as follows:

1. Passively relocating the FOWTs in a FWF increases the farm's energy production.

2. Allowing larger FOWTs excursions decreases the fatigue loads on the mooring lines.

3. Allowing larger FOWTs excursions decreases the extreme loads on the mooring lines.

4. Fatigue loads on a mooring line are highest when the mooring line is upwind and aligned with the wind direction.





The paper first introduced the four steps of the FWF layout design methodology and showed the results of each step using static models. The results of the static models showed that at a constant wind speed of 10 m/s, FOWT movements can increase the energy production of the FWF by 3.1%. Afterward, we verified these results using the polar implementation of the DWM model within FAST.Farm. First, we simulated the baseline FWF with all FOWTs coupled to the baseline mooring system, which restricted the FOWTs' movements. Then we simulated the FWF with each FOWT coupled to the customized mooring system design allowing crosswind movements to mitigate the wake losses. This was repeated for all wind directions of the wind rose presented in Fig. 2. Finally, we compared the results of the energy production of the baseline FWF to the new FWF layout design. The energy gain calculated by FAST.Farm was 1.4% at a constant wind speed of 10 m/s; this is 45% of the energy gain predicted by the steady-state wake models. However, the results showed that FAST.Farm overpredicts the vertical wake deflection for FOWTs and hence underestimates the wake losses within the baseline FWF layout. This causes the big difference between the steady-state wake model and the DWM model as explained in Section 3.4. Calibration of the DWM model within FAST.Farm using higher fidelity large-eddy simulations are needed to decrease the uncertainty in the wake losses inside an FWF.

The study also assessed the performance of the novel mooring system designs and their fatigue loads in operational conditions. Wind and wave simulations were conducted at rated wind speed of 11 m/s for all wind direction with an increment of 10° (i.e., 36 wind directions). Then we calculated the 1 Hz DEL and the 25-year damage. The results showed that the customized mooring system designs have lower 25-year damage than a state-of-the-art mooring system design with the same mooring line diameter. This is due to the lower stiffness of the customized mooring system designs, which leads to smaller dynamic fluctuations in the fairlead tensions. Moreover, the results showed a direct relationship between the line headings, wind direction, and DEL values. The fatigue loads were highest when the wind was aligned with the upwind mooring line. Therefore, in any FWF we should avoid aligning the mooring lines with the most probable wind direction in order to avoid having high fatigue loads concentrated at one of the mooring lines.

Furthermore, we checked that the customized mooring system designs can endure 50-year extreme wind and wave events for all 36 wind directions with an increment of 10°. The results showed that the maximum loads on the new customized mooring system designs were always lower than a baseline state-of-the-art mooring system design. All customized mooring system designs comfortably withstood extreme 50-year events for all wind directions when applying a safety factor of 2.

In conclusion, this study successfully verified the potential of relocating FOWTs to increase energy production within FWFs using a mid-fidelity dynamic analysis, which is the primary goal of this research. Despite the uncertainties associated with the DWM model for wake losses, the dynamic results predicted an energy gain of 1.4% at a constant wind speed of 10 m/s. Additionally, the FOWTs moved as predicted in the design process and the energy gain distribution in Fig. 11 was as predicted by the steady-state model in the design process. Additionally, the paper showed that designing mooring systems with larger excursion limits and lower stiffness leads to lower mooring system fatigue damage and extreme loads. This means that allowing larger excursions leads to mooring lines with smaller diameters. In future work, we are planning to make our designs more realistic and consider real-life scenarios. Therefore, we are planning to include the levelized cost of energy as a design objective along with increasing the energy gain. Moreover, we are planning to consider the supply chain and ease of installation





by designing a single mooring system for all FOWTs in the FWF instead of having a different mooring system design for each FOWT in the FWF.

*Code and data availability.* The code used for FWF layout optimization, and the MoorDyn file for each mooring system design can be found here https://doi.org/10.5281/zenodo.8370977

*Video supplement.* An animation video of the passive displacements of the FOWTs can be found here http://doi.org/10.5446/63167.

## Appendix A: FWF mooring system designs

**Table A1.** Customized mooring system designs

| T | Line 1 | | | | Line2 | | | | Line 3 | | | |
|---|--------|--------|--------|--------|---------|--------|--------|--------|---------|--------|--------|--------|
| | Heading | d | Anc. r | L | Heading | d | Anc r | L | Heading | d | Anc r | L |
| | [°] | [m] | [m] | [m] | [°] | [m] | [m] | [m] | [°] | [m] | [m] | [m] |
| 1 | 30 | 0.15 | 840 | 944.91 | 170 | 0.15 | 840 | 909.85 | 280 | 0.15 | 840 | 944.91 |
| 2 | 210 | 0.15 | 840 | 909.85 | 0 | 0.15 | 840 | 909.85 | 100 | 0.15 | 600 | 707.94 |
| 3 | 215 | 0.15 | 840 | 944.91 | 335 | 0.15 | 840 | 944.91 | 95 | 0.15 | 840 | 944.91 |
| 4 | 80 | 0.15 | 840 | 909.85 | 210 | 0.09 | 840 | 909.85 | 350 | 0.12 | 600 | 674.9 |
| 5 | 240 | 0.09 | 840 | 944.91 | 10 | 0.15 | 840 | 909.85 | 140 | 0.12 | 840 | 909.85 |
| 6 | 270 | 0.12 | 840 | 944.91 | 30 | 0.15 | 840 | 944.91 | 160 | 0.15 | 840 | 909.85 |
| 7 | 265 | 0.12 | 840 | 909.85 | 25 | 0.15 | 840 | 909.85 | 165 | 0.15 | 600 | 674.9 |
| 8 | 70 | 0.15 | 600 | 674.9 | 200 | 0.12 | 840 | 909.85 | 340 | 0.15 | 840 | 909.85 |
| 9 | 190 | 0.15 | 840 | 909.85 | 340 | 0.15 | 840 | 909.85 | 80 | 0.15 | 600 | 707.94 |

*Author contributions.* MYM developed the novel method to passively relocate the FOWTs in the FWF, carried out all the static and dynamic simulations, and wrote most of the paper. EL designed the baseline mooring system, contributed to the fatigue analysis post-processing and helped in writing and reviewing the paper. MH contributed to the conceptualization of the methods and reviewed the paper. PWC supervized
the work and reviewed the paper.

*Competing interests.* The authors declare that they have no conflict of interest.



*Acknowledgements.* The support of Jason Jonkman, Andrew Platt and Lucas Carmo in setting up the FWF model in FAST.Farm is gratefully acknowledged. The research leading to these results has received partial funding from the European Union's Horizon2020 research and innovation programme under grant agreement no.815083 (COREWIND). This work was authored in part by the National Renewable Energy Laboratory, operated by Alliance for Sustainable Energy, LLC, for the U.S. Department of Energy (DOE) under Contract No. DE-AC36-08GO28308. Funding provided by U.S. Department of Energy Office of Energy Efficiency and Renewable Energy Wind Energy Technologies Office. The views expressed in the article do not necessarily represent the views of the DOE or the U.S. Government. The U.S. Government retains and the publisher, by accepting the article for publication, acknowledges that the U.S. Government retains a nonexclusive, paid-up, irrevocable, worldwide license to publish or reproduce the published form of this work, or allow others to do so, for U.S. Government purposes.



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
