# Peer review of "Dynamic performance of a passively self-adjusting floating wind farm layout to increase the annual energy production"

_Wind Energy Science, 2023_

## Author Comment (AC1)

**Authors' response to reviewer 1**

We thank the reviewer for the valuable comments and suggestions, which we consider very important and help us to sharpen and improve the manuscript. Here are our responses written in green to each comment.

The authors response is shown in green.

**General comments:**

It is suggested to shorten the paper to focusing on the main results only as in the current form it is hard to follow and read.

Thank you for your suggestion. We realised that the paper is long while writing it and we actually kept it as short as we can and we even decided not to add some sections to avoid making it longer. We believe removing any of the current sections takes away from the goal of the paper and its structure. The paper mainly aims on showing three main points:

- The results of the relocating FWF layout design using steady state wake models (Gaussian wake model in FLORIS) and static tools for mooring systems (using MoorPy). This can only be done through a quick introduction of the method and the results obtained at each step. Showing only the final results when using steady state tools will add ambiguity to the paper, and makes it dependent on our previous work and cannot be read as a standalone paper.

- The dynamic results of the same relocating FWF using OpenFAST and FAST.Farm.

- The comparison between the steady state results and the dynamic results.

Removing any of the sections of the paper will take away from the paper's integrity and our ability to fully present these three points. However, we decreased the level of details in section 2.3, and made it shorter.

It is suggested to change the writing style from the first to the third person

Thank you for your comment. We know there is a debate on what writing style to use. However, it was suggested for us by the internal editor at NREL to use first person, so we prefer to keep using it in this work.

The power gain increase achieved by the authors is insignificant, so it is hard to justify the application of the proposed mooring system design to industry

The energy gain of the new design should be compared to the losses in the baseline layout design and not as an absolute value. A system with small losses will have small gain after optimization, while a system with higher losses should have higher gain. This was missing in the paper but now we added tables 3 and 4 to show it, which we will mention also later in the response.

Moreover, the paper is presenting a preliminary design method and more work needs to be done before proposing this method as mature for industrial application (For example, we do not consider the cable design in this work) to include large lateral displacements of the FOWT. We are also aware that the energy gain will even decrease more when the full wind rose is applied as no gain will be produced form the above rated wind speeds, and less gain will be produced for lower wind speeds.

The paper introduces a method to integrate the mooring system design as part of the FOWT's design. The paper shows that these movements are capable of affecting the AEP of the wind farm. Therefore, we do not think that the gain presented in this work is insignificant and we will go through this in detail while discussing the last comment to avoid redundancy.

**Technical comments:**

Line 30, the statement "... provides stiffness in surge, sway, and yaw degrees of freedom" is only valid for slack-mooring and is not valid for TLPs

Thank you. We updated the text as shown below:

"For a FOWT, the mooring system is responsible for station-keeping, as the catenary mooring system provides stiffness in surge, sway, and yaw degrees of freedom (DoFs)."

Line 110, the authors demonstrate a gain of 6.1% at 10 m/s, it would be good to know the power gain using the entire wind probability at a particular deployment location

Thank you for your comment. We believe calculating the full wind rose energy gain lies out of the scope of this paper for the following reasons:

- The goal of the paper is to verify the results of steady states models to the dynamic models. Therefore showing the maximum targeted gain for a full wind rose is out of scope of this work and was presented in our work in [1].

- The wind rose used is a theoretical wind rose created within IEA Task 37. Therefore, we cannot apply any wind rose for a location of deployment as this means we need to redesign the FWF and the mooring system for this new wind rose.

We updated now the text in section 2.2 as follows: "The energy gain achieved in through this paper only considers a constant wind speed of 10 m/s, and not the full wind spectrum. As shown in our work in [1], when all wind speeds are considered the energy gain was reduced by 40% to 30% of the gain calculated at 10 m/s. The calculation of the full wind rose energy gain is out of scope of this paper because we instead focus on the comparison of the steady state and dynamic models."

Figure 8 and 9 - use the power of 10 to show Frequency of the FOWT, and the chosen color-scheme does not demonstrate the variation in frequency range

Thank you for your suggestion. We updated the Figures to show the frequency as a power of 10 as suggested. We have also changed the color-scheme as suggested.

Figure 11 is mentioned first on page 6 while it appears on page 14

Yes, this is true. We preferred to show this Figure as a comparison rather than splitting it into three figures. We believe this allows the reader to compare the results at different stages of the process easier. However, we removed the early mention of Figure 11.

Section 3.2 - the authors refer to the natural frequency of FOWT but not clear which DOF

Thank you for the clarification. We used x-axis and y-axis in Figures 8 and 9. We will update the text to include that these are for surge and sway DoFs. We updated the captions of the figures and line 224 is now updated as follows:

"The figures show the value of the natural frequencies in the $x$-axis and $y$-axis directions (surge and sway DoFs respectively)."

Figure 8 and line 230 - the authors state that the natural frequency does not change with the wind speed and wind direction while in reality it is. It has been shown in https://asmedigitalcollection.asme.org/OMAE/proceedings-abstract/OMAE2023/86908/1167328 that the natural frequency in surge changes with wave and wind directions

Thank you for the clarification. This is true, as we indicated in Figures 8 and 9, the stiffness is changing with wind speed and direction. In line 230 we say that for the baseline mooring system design the stiffness is almost constant for all wind speeds and all wind directions in comparison to the new less stiff mooring systems which have bigger changes with wind speed and direction. We understand that our explanation for this was unclear and confusing and updated the text as shown below:

"In Fig. 8, the natural frequency of the baseline design shows only small changes as the wind speed and wind direction change. This means that for all wind excitations and for all positions inside the watch circle, the natural frequency is changing within a small range as shown in Fig. 8. This is because the stiffness of the baseline design is linear and almost constant for all wind speeds and wind directions."

Line 265 - the choice of the sea state parameters should be explained

Thank you for pointing this out. We updated the text as shown below:

"The significant wave height for all simulations was equal to 2 m, and the wave period was equal to 6 s. We used these values for the sea states as they are the operational values used during the Activefloat design as indicated in [2]."

Line 275 - the gain of 1.4% might be within the modelling error and is insignificant

Thank you for this comment. We have added context to explain why the 1.4% gain is significant.

- The 1.4% is a small value when we do not consider the value of the wake losses. However as shown in Table 4 newly added to the text the new design leads to decreasing the wake losses from -6.08% to -4.73%. This decrease is equivalent to a decrease of wake losses by 22%. The gain in our work is only 1.4% because we used the optimized wind farm layout OWFL as a baseline for comparison. We decided to do this instead of using a gridded shape layout to truly show the benefit of relocating the FOWT. Starting with a gridded shape layout would lead to a much higher gain value as the gridded layout has higher losses. We are currently working on a paper showing that for wind farm layouts that are gridded shaped similar to the Horns Rev I wind farm, this method has a much higher potential as any small relocation will significantly increase the power. Tables 3 and 4 are now added to the text to show the wake losses of each layout.

- There is a big uncertainty from the wake model implemented within FAST.Farm as we discuss in the paper. However, FAST.Farm currently underestimates the wake losses for a FWF. As we discussed in the paper the work done in [3], compared the FAST.Farm results to MIRAS-HAWC2 results. The results show that FAST.Farm over estimates the vertical deflection of the wakes due to the pitching of the FOWT. This overestimation decreases the wake losses predicted by FAST.Farm and this is explained in section 3.4. Therefore, the uncertainty in the wake model decreases the

[Figure]

Figure 1: Final energy gain in FLORIS on the left and in FAST.Farm on the right

energy gain due to relocating the FOWTs and does not increase it.

- In Figure 11, that we zoom in and introduce again here, the final gain expected from the static model and the gain from FAST.Farm follow the same pattern. If the gain was due to numerical uncertainty the gain distribution over the wind directions will be random and would not agree with the predictions of the static model. This is explained in the paper in section 3.4 in line 290. The text is now updated as follows: "The energy gain distribution per wind direction in the FAST.Farm model follows the same trend as the gain distribution for the MoorPy-FLORIS results. This shows that the energy gain achieved by the OWFL when coupled to the OWFL, is not numerical as it is not random but follows our expectations from the steady state model."

- In section 3.3 we showed that the lateral movements of the FOWTs and OpenFAST match each other. This minimizes the probability of the gain being a numerical modelling error.

**References**

[1] Mohammad Youssef Mahfouz and Po Wen Cheng. A passively self-adjusting floating wind farm layout to increase the annual energy production. *Wind Energy*, 26(3):251–265, 2023.

[2] Mohammad Youssef Mahfouz, Mohammad Salari, Fernando Vigara, Sergio Hernandez, Climent Molins, Pau Trubat, Henrik Bredmose, and Antonio Pegalajar-Jurado. D1.3. Public design and FAST models of the two 15MW floater-turbine concepts, December 2020. This deliverable is a draft version, and still under revision by the EC.

[3] Néstor Ramos-García, Sergio González-horcas, Antonio Pegalajar-jurado, Ozan Gözcü, Henrik Bredmose, Umut Özinan, Mohammad Youssef Mahfouz, Alessandro Fontanella, Alan Facchinetti, and Marco Belloli. D1 . 5 : Methods for nonlinear wave forcing and wakes. Technical Report March 2022, 2023.

---

## Author Comment (AC2)

**Authors' response to reviewer 2**

We thank the reviewer for the valuable comments and suggestions, which we consider very important and help us to sharpen and improve the manuscript. Here are our responses written in green to each comment.

The authors response is shown in green.

**General comments:**

The paper describes a methodology to design mooring systems that passively adjusts the position of floating wind turbines in a farm to avoid wakes from upwind turbines. The thinking is innovative, and the authors seek to demonstrate the energy yield gains in applying the methodology. The paper is well written and very detailed, perhaps too detailed, as it is on the long side and sometimes difficult to follow. The discussions of the observed differences in results between FAST/MoorPy, FAST.Farm/FLORIS are important and clarifying. However, descriptions of these two could be presented in a way that gives a better overview for the reader that is not familiar with these softwares.

Thank you for your kind comment. We tried to be detailed to help the reader understand the methods presented. This is a new topic with a new method introduced and we wanted to increase the level of clarity of our work. We agree that following the paper can be hard without understanding the tools used, but these tools are well introduced in details in many references. We have cited all of these references in detail for the reader. Since the development of these tools is not part of our work, we believe that more details on the tools is out of scope of the paper.

It is also unclear why 10 m/s was chosen. Something about at which wind speeds the highest gains are expected should be mentioned. Also, it is mentioned in the conclusions that "a more realistic" case will be studied in the future, but it would be useful to know at this point what the design strategy would be if one had to design mooring systems for more than one wind speed. It would also be interesting to know the author's thoughts on how significant the observed increase in energy yield is for the total AEP.

Thank you for your comment. The methodology is independent of the wind speed value. We chose 10 m/s because this is where we predict the gain to be the highest as it is the wind speed just below the rated wind speed of the FOWT at 11 m/s. We cannot use above rated wind speeds to design the mooring systems in this method, as above rated there are no power losses due to the wakes. We updated now section 2 in the text as follows "The 10 m/s wind speed used has no effect on the FWF design because the method is independent of the wind speed.. A wind speed just below the rated wind speed of the turbine is recommended as this is where the energy gain is the highest."

When referring to a "more realistic" case in the conclusion, we meant that we would add more constraints on the same method. For example, we are now

writing a paper about applying the method on the Horns Rev I wind farm layout, while choosing only one mooring system design for all FOWTs in the farm and not a different mooring system for each FOWT. Moreover, we are also planning to decrease the maximum displacement done by the FOWT. We updated the text now as follows: "We are planning to use one customised mooring system design for all FOWTs in the FWF instead of having different mooring system for each FOWT. Additionally, we will decrease the FOWTs excursion limits to bring them closer to the current limits used within the current state-of-the-art mooring system designs."

The energy gain will be lower when the full wind rose is applied as no gain will be produced at above rated wind speeds, and less gain will be produced for lower wind speeds. In our work in [1], the gain was reduced by 40% to 30% when a full wind rose was applied. We did not want to add it in this work because the goal here is to study the dynamic performance of the customised MS design and the FWF power production. Section 2.2 is now updated as follows "The energy gain achieved in through this paper only considers a constant wind speed of 10 m/s, and not the full wind spectrum. As shown in our work in [1], when all wind speeds are considered the energy gain was reduced by 40% to 30% of the gain calculated at 10 m/s. The calculation of the full wind rose energy gain is out of scope of this paper because we instead focus on the comparison of the steady state and dynamic models."

The description of the mooring system database in sec 2.3 is very detailed, and quite confusing. I understand that the concepts of "mooring system watch circle" is described in a previous paper, but it would be helpful if the concept was described better in the current paper.

Thank you for your comment. We have restructured section 2.3 and decreased the level of detail. We have now added a paragraph and a Figure to explain the watch circles..

It is stated that the allowable displacement of the mooring system is 1D. For a 15MW turbine at 200 m water depth, this is 120% of the water depth. Normal offset requirements to secure cable integrity is in the range of 10-30% of the water depth. This criterion is also why one ends up with fairly stiff mooring systems. Soft systems like the one in the design here, also could have other issues that are not addressed here, such as snap loads. Please comment on this.

Thank you for bringing this up. We are aware that currently there is a constraint on the excursion done by the FOWT due to the cable design. However, to the best of our knowledge we have not see any work done discussing the limits of the cable or studying the effects of such motions on the cable's fatigue loads. We think that these limitations come from the oil and gas industry and more research is needed to understand what are the real physical limits. We hope our work will motivate more research in this direction, to answer this question regarding the cable. We updated the text in section 2.3 as follows: "Although the current state of the art generally limits the FOWTs excursions to be less

than 30% of the water depth, we are going to neglect this in our current work, and allow larger excursion limits."

Moreover, even with 30% displacement if we integrate the mooring system design as part of the FWF layout design we can benefit from these motions to increase the AEP of the farm even a small increase in AEP can be valuable. This is one of the aspects we are planning to look into in our next paper and this is what we meant by more realistic design in the conclusion. We updated the text in the conclusion as follows: "In future work, we are planning to make our designs more realistic and consider real-life scenarios. We are planning to use one customised mooring system design for all FOWTs in the FWF instead of having different mooring system for each FOWT. Additionally, we will decrease the FOWTs excursion limits to bring them closer to the current limits used within the current state-of-the-art mooring system designs."

Finally, you are correct that there are other issues to look after while designing the soft mooring systems. We have checked for snap loads in all our simulations for fatigue and extreme loadings. There were no incidents of snap loads in any of the customised mooring system design. We believe this is because of the constraints we had on the mooring system database results where very soft mooring systems where not accepted by the constraint on the maximum allowable yaw angle. We updated sections 3.5 and 3.6 as follows: "Finally, we checked the mooring lines tensions for snap loads as the mooring systems presented are less stiff than the state of the art mooring designs. However, we have not seen any snap loads in the operation conditions in any of the cases we checked."

"Finally, we have not seen any snap loads in the extreme loading conditions in any of the cases we checked."

Continuing on soft mooring systems. It is weel established that soft mooring systems experience less fatigue than stiff systems, thus the difference in fatigue damage between the base case and the adjusted system is not necessarily related to the fact that it uses passive position adjustment. It is therefore not fair to compare the fatigue damage to a base case that was designed for a 12%WD offset.

Thanks again for bringing this up. Yes, we also expected the soft mooring system designs to have lower fatigue loads. However, we believe this comparison is valuable for two main reasons. First, it shows that allowing bigger motions of the FOWTs decreases the fatigue loads on the mooring system, which means we can use smaller mooring diameters can be used to achieve the same fatigue damage. It is an advantage for the new customised mooring systems over any of the current state of the art mooring system designs that should be highlighted.

Second, the main goal of this paper is to study the dynamics of the customised mooring systems design, and a crucial part of this is the fatigue response.

**Specific comments:**

Sec 2.4: What is "brute force optimization"?

Brute force optimization is an optimization in which all possible solutions are tried. We did not explain in detail the optimization process in this paper. This is the core part of our method and is explained in details in our work in [1]

Sec 3.2: It is stated that the base case design is a linear mooring system. Is it not catenary? Please explain.

It is stated that the stiffness is linear, not that the mooring system is linear. This means that the force-displacement curve is linear and the stiffness does not have big changes as the force acting on the FOWT changes..

Fig 8 and 9: It would be helpful if the text transfers these frequency ranges to periods.

The captions are now updated as follows "The range of the colour bar extends from 0.001 Hz (time period of 900 s) to 0.01 Hz (time period of 90 s).".

Sec 3.3: Are tower top deflections really that significant for platform offset, compared to (the mentioned) platform rigid body motions?

In this section, we are stating the difference in the MoorPy model and the OpenFAST model. The MoorPy model is static does not consider any dynamic effects, while the OpenFAST model is a time domain dynamic model. It is expected that there will be small differences between the two models as there is a difference in the details used to model the turbine's structure and excitation forces. In this part we are stating the difference, leading to the small deviations in the watch circle of both models. However, we do not think that the tower deformation has a higher effect, we believe the difference comes from the change in the aerodynamic forces acting on the rotor as the floating platform moves in pitch, roll, and yaw. This is well captured by the BEM model in OpenFAST but not in MoorPy. However, to be able to quantify this we would have to simulate a case while turning off the DoFs of the tower in OpenFAST to make it rigid.

Fig 11: Please include units in these figures.

There are no units in this figure as it shows the percentage energy gain calculated using equation 1. We updated the caption as follows "(left) The percentage of energy gain at each wind direction for the targeted wind farm layout, (center) the percentage of the final wind farm layout using static wake models, and (right) the percentage of the final wind farm layout using dynamic wake models.".

**References**

[1] Mohammad Youssef Mahfouz and Po Wen Cheng. A passively self-adjusting floating wind farm layout to increase the annual energy production. *Wind Energy*, 26(3):251–265, 2023.

---

## Author Comment (AC3)

**Authors' additional response**

The cost of the mooring designs was not included in the paper first version. However, allowing a larger displacement of the FOWT lead to a less stiff mooring designs. A less stiff catenary mooring design will have a smaller mass and hence the material costs of the mooring systems will decreases. This will lead to a decrease in levelized cost of energy (LCOE) of the floating wind farm. According to [1], the cost of the mooring lines can be split into two parts, the chain costs and the drag embedded anchor (DEA) costs as shown in equations 1, and 2 respectively. All mooring lines are assumed to have the same drag embedded anchor design. The minimum breaking load (MBL) is calculated using equation 5 from the paper, and $L$ is the mooring line length. The cost in equations 1, and 2 is calculated in US dollars.

$$Chain_{cost} = (0.0591 \cdot MBL \cdot 10^{-3} - 87.6)L \tag{1}$$

$$DEA_{cost} = 10.198 \cdot MBL \cdot 10^{-3} \tag{2}$$

Using this equation the material cost of the baseline mooring system design is 38.5 million dollars for the entire wind farm, while the cost of the new customised mooring system designs is 30.7 million dollars. This means that the material costs of the new mooring system design is 20% less than the baseline mooring system design. The details of the customised and baseline mooring systems design parameters are shown in Table A1 and Table 2 in the paper respectively.

This shows that in addition to steering the wake and increasing the farm energy production, a more flexible mooring system design is cheaper and hence decreases the LCOE of the wind farm even further. If the editor and the reviewers would find this section relevant and informative, we will be happy to add it to the paper to emphasise the potential of considering a softer mooring system design to decrease the LCOE while designing the mooring system.

**References**

[1] Philipp Beiter, Walter Musial, Aaron Smith, Levi Kilcher, Rick Damiani, Michael Maness, Senu Sirnivas, Tyler Stehly, Vahan Gevorgian, Meghan Mooney, and George Scott. A Spatial-Economic Cost- Reduction Pathway Analysis for U.S. Offshore Wind Energy Development from 2015–2030. *National Renewable Energy Laboratory (NREL)*, (September):214, 2016.

---

## Author Response (AR2)

**Authors' response to reviewer 3**

We thank the reviewer for the valuable comments and suggestions, which we consider very important and help us to sharpen and improve the manuscript. Here are our responses written in green to each comment.

The authors response is shown in green.

**General comments:**

The present work summarizes the previously published approach for optimization of a passively adjusting floating wind farm and examines 1) the application of DWM rather than static methods and 2) the dynamic responses of the FWTs. The results are believable and the work is generally understandable, and the results are basically as expected (allowing larger mean deflections gives a softer mooring system, which takes up fewer dynamic loads than a stiffer system and thus appears to perform "better" than a "conventional" system).

Thank you.

In general, it would be good to include some more caveats about the results. For example, claiming improved fatigue life based on only simulations of a single environmental condition is a bit misleading. The fact that the baseline design allows for a conventional power cable design, while the passive repositioning system requires a power cable that can tolerate much larger offsets, should also be highlighted. I would furthermore argue that a chain-only design is not exactly state-of-the-art for floating wind, where lighter systems with lower environmental impact are also being examined.

These are all very valuable and useful comments and we would like to thank the reviewer. To consider the reviewers recommendation the following was done:

- We went through the paper and emphasised in the abstract, in section 3.5, and in the conclusion that the results shown for fatigue damage are non conclusive for the overall lifetime fatigue damage of the FOWT and only represent the loads at rated wins speed.

- We updated the text below Figure 8 as follows: "The baseline design can be coupled with a conventional power cable design, while the passive repositioning mooring system designs require a power cable that can tolerate much larger offsets. The effect of the larger FOWT's relocation on the cable design is not discussed in this paper.". Additionally, the following sentence was added to the conclusion "Additionally, studying the effect of the FOWT's larger displacement on the cable costs and cable design is still missing in the current work."

- To address the material comment we added the following sentence to the text "Novel lighter materials with lower environmental impact for the

mooring lines are not checked in this work."

When examining the dynamics of the system, it is also very relevant to discuss the control system applied in the simulations. Most reference control systems are actually unstable in surge, and would in principle be "more unstable" for longer natural periods. Did you deal with this issue, or simply ignore it?

The controller used in this work was tuned for the baseline Activefloat platform as shown here [1, 2]. There was no retuning for the controller within this work. We did not see any instabilities with any of the mooring system designs in any of the dynamic simulations we did in the current work. We added this paragraph to the text "The controller tuned for the Activefloat coupled to the 15 MW reference turbine in the work of [1, 2], is used in the current work. The controller is a simple generator torque controller for the below rated wind speed and collective pitch controller for the above rated wind speeds with no tower top feedback and no yaw control. The controller was not retuned in this work with the change of the mooring system designs and was used as is for all simulations in this work."

The issues related to wake deflection prediction are quite interesting. It is possible to tune the FAST.Farm parameters to achieve similar wake deflection as LES simulations for this turbine, as shown in `https://doi.org/10.1016/j.renene.2023.119807`. On the other hand, it wasn't possible to match both yaw and tilt-induced deflections simultaneously. I would also note that DWM may not accurately model the boundary flow around the farm very accurately as the number of turbines increase (especially with pitched floaters causing upward wake deflections).

Yes, they are very interesting. I was not aware of the paper shared by the reviewer. Thank you for sharing it as this is very useful to my work and my future research plans. I have checked it and it adds credibility to the results. Therefore I added the reference to the conclusion section.

**Specific comments:**

Line 26: why does "open" need to be specified here?

There is no need to specify the tools being open-access here. It is removed to avoid confusion. Thank you for catching this up.

Line 92-93: Is a minimum distance constraint actually needed? Why doesn't the optimizer capture the fact that larger losses occur and avoid this area anyhow?

Thank you for your question. The minimum distance is needed especially for the less probable wind directions of the wind rose. The gain for getting

two turbines really close together (less than the minimum distance) in the less probable wind direction can lead to higher energy gain for more probable wind direction. Therefore, setting the minimum constraint is crucial to avoid two turbines closer than the minimum defined distance..

Line 102: wouldn't it be even better to design a passively adjusting layout from scratch, rather than assuming that it is simply a correction to the optimized fixed layout? This is probably too computationally expensive in practice, but I don't see a reason to believe that the results would be identical.

That is a very good comment. Yes, we do not expect the results to be identical if a passively adjusting layout is designed from scratch. We did not go for this approach cause our goal now is to proof that there is a potential to having these soft mooring lines and allowing passively adjusting layouts. We are not claiming that the design presented in this paper is the global optimum. We have updated the introduction to clarify this "The wind farm layout and the mooring system designs presented at this work are not the global optimum designs, as this is not the goal of the method as stated and presented in our work [3]."

Moreover, we prefered to follow the step by step approach we used to show the targeted energy gain which can be achieved. This energy gain represents the limit of the optimization process as shown in Table 2 it decreases the wake losses by 2%, such targeted layout cannot be emphasised if an optimization process from scratch was used.

Finally, we updated the conclusion to include this point as future research as we believe this should be the future of floating wind farms designs. This sentence is now added to the conclusion "Finally, the optimization process should include from the beginning the ability of the FOWTs to relocate their positions passively, and this should be integrated in a full optimization routine instead of the step by step approach followed in this paper."

Line 125: I'm having a hard time following "the energy gain was reduced by 40% to 30% of the gain calculated at 10 m/s." Is $30 - 40\%$ a range, or do these percentages refer to two different things?

Thank you for the comment. The results discussed in the paper [3], showed the AEP for two different wind roses with identical wind direction probability distribution, but different Weibull distribution at each wind direction. This led to a difference in AEP gain. We updated now the text as follows: "The 40% to 30% values represents the results of two wind roses with identical wind direction distribution but different Weibull distribution."

Line 168: Do I understand correctly that the yaw angle is calculated, but nonetheless not included in FLORIS?

Yes, this is correct. In FLORIS, the pitch, roll and yaw angles of the FOWTs are not included. This is also mentioned previously in line 99 therefore we did not update the text here.

I like the concept of figures 9 and 10, but have a hard time reading actually frequencies with these colors. Would it be possible to give a range of natural periods along the 10 m/s watch circle in the text? This would be useful in understanding, for example, how different we expect the behavior to be, or whether or not 3600 s is sufficient to capture a sufficient number of oscillations at the natural frequency.

We added the range of natural periods to the text as requested by the reviewer. The text added is the following "As an example to the spread of the natural frequency of the customised mooring systems at wind speed of 10 m/s, the baseline mooring system design natural periods range from 97 s to 130 s in the x-axis direction, and ranges from 97 s to 147 s in the y-axis direction. On the other hand at 10 m/s the natural periods of the customized mooring system design coupled to turbine 1 ranges from 108 s to 404 s in the x-axis, and from 128 s to 448 s in the y-axis. The dark areas in Figures 9, and 10 represents the higher natural period region (lower natural frequency). The x-axis and y-axis do not change with the change of wind direction and are fixed global axis."

Line 292: "OWFL when coupled to the OWFL" – I understand what you mean, but could be worded better.

Thank you for catching this typo. We updated the sentence as follows "This shows that the energy gain achieved by the OWFL when coupled to the customised mooring systems,..... "

Line 315: Is the rotor-average wind speed defined in rotor coordinates?

Yes, this is true and thank you for the question. However, the rotor coordinates are not identical in both tools. The rotor is tilted and pitched in FAST.Farm while this is not the case in FLORIS where the rotor is not tilted or pitched. This is explained in the text in lines between 317 to 320. We added this sentence to clarify "This is because the rotor average wind speed is measured in rotor coordinates for both tools, but the rotor coordinates are fixed in FLORIS while they move with the rotor in FAST.Farm."

Line 320: when discussing wake meandering, it is confusing to me whether you refer to the deficit part of the model, deflection part, meandering part, or all 3.

I am discussing all three parts of the model. I have went throw the paper and updated the text to make sure I am using the correct verb for each part. When talking about meandering I used the verb meander, for deflaction I used deflect and for the deficit I referred to it using wind speed.

Line 328: How large is the effect of the mooring system on the mean pitch angles?

The effect on the mooring system on the pitch angle is negligible and less than 1°. It depends on the overhang of the mooring lines and which line is heavier and if it is upwind and downwind. This is why the change in wind

speed is also negligible less than 0.1 m/s. The text is updated as follows "The change in the mean pitch angle is small and less than 1°."

Line 353: It would be interesting to know whether the dynamic motions or just the mean motions matter in the comparison. You could run the baseline model with the tilt corresponding to the tilt+pitch of the floater and a yaw angle equal to the mean yaw angle to see this.

Yes, this is an interesting point. In this work, we already updated the FAST.Farm source code (AeroDyn module) to remove the dynamics of the motions of the floater while calculating the rotor average wind speed. Therefore, we believe that all the results we show in this paper in all cases of FAST.Farm do not have an effect of the FOWTs dynamics but only the mean offsets are considered. Therefore, a more detailed study is out of scope of our current work. We updated the text as follows (Line 316): "We updated the source code of the FAST.Farm used in this work so the rotor average wind speed do not include the FOWTs dynamic motions."

Line 358: wouldn't it be easier to write 0.17D?
You are right thank you. We updated the text as suggested.

Figure 15: Check figure title
Thank you for catching this. The Figure is updated.

Line 375 and 397: were/where
Thank you. We updated the typos mentioned.

Line 413: note that the reduced stiffness will in principle reduce the platform motion responses to first-order wave forces (as the system is even farther from resonance). This difference is probably small in this case, though.

Thank you so much. We have added this comment to the text as we believe it adds a lot of value.

**References**

[1] Mohammad Youssef Mahfouz, Climent Molins, Pau Trubat, Sergio Hernández, Fernando Vigara, Antonio Pegalajar-Jurado, Henrik Bredmose, and Mohammad Salari. Response of the International Energy Agency (IEA) Wind 15 MW WindCrete and Activefloat floating wind turbines to wind and second-order waves. *Wind Energy Science*, 6(3):867–873, 2021.

[2] Mohammad Youssef Mahfouz, Mohammad Salari, Fernando Vigara, Sergio Hernandez, Climent Molins, Pau Trubat, Henrik Bredmose, and Antonio Pegalajar-Jurado. D1.3. Public design and FAST models of the two 15MW

floater-turbine concepts, December 2020. This deliverable is a draft version, and still under revision by the EC.

[3] Mohammad Youssef Mahfouz and Po Wen Cheng. A passively self-adjusting floating wind farm layout to increase the annual energy production. *Wind Energy*, 26(3):251–265, 2023.

---

## Author Response (AR3)

**Authors' second response to reviewer 2**

We thank the reviewer for taking the time to review the paper. We also thank the editor team for their time and patience throughout the review process. We would like to emphasize that the missing reply was not out of negligence and that we respect how the review process increased the quality and readability of the paper. The reason that we did not provide a reply earlier is that most of our following replies are redundant and almost identical to our replies to the reviewer the first time. Here are our responses written in green to each comment.

The authors response is shown in green.

**General comments:**

Although this paper presents an interesting and innovative study, which I think deserves publication, the presentation of the study falls through. The authors fail to take the feedback in the previous round properly into account. In particular on the two following issues:

We thank the reviewer for his opinion about the novelty and the innovative topic. We are sorry if the reviewer did not feel that our replies met his expectations, as we worked hard to cover all the points they brought up in the earlier round of review.

- Both reviewers recommended to shorten and focus the paper in the previous round. However, this comment has not been taken into account.

In the first review, the reviewer said the paper was well written, but some sections were hard to follow. Hence, the authors have attempted to condense the information to increase the readability of the study, especially for the section the reviewer mentioned in the first feedback.

The reviewer specifically mentioned section 2.3 in the following quote:

*"The description of the mooring system database in sec 2.3 is very detailed, and quite confusing. I understand that the concepts of "mooring system watch circle" is described in a previous paper, but it would be helpful if the concept was described better in the current paper."*

Therefore, we restructured the entire 2.3 section as recommended by the reviewer in the revised manuscript and added Figure 6 in the revised version for the watch circles to increase clarity.

We realized that the paper was long while writing it, and we kept it as short as we could. We even decided not to add some sections to avoid making it longer. We believe removing any of the current sections takes away from the paper's goal and its structure. The paper mainly aims to show three main points:

- The results of the relocating FWF layout design using steady-state wake models (Gaussian wake model in FLORIS) and static tools for mooring

systems (using MoorPy). This can only be done through a quick introduction of the method and the results obtained at each step. Showing only the final results when using steady-state tools will add ambiguity to the paper, make it dependent on our previous work, and cannot be read as a standalone paper.

- The dynamic results of the same relocating FWF using OpenFAST and FAST.Farm.

- The comparison between the steady state results and the dynamic results.

Removing any of the sections of the paper will take away from the paper's integrity and our ability to fully present these three points. However, we decreased the details in section 2.3 and made it shorter.

In short, the authors had to choose between clarity and concision, and in some cases, one affected the other

- On the research methodology: The study compares a soft mooring system with passive layout adjustment to a stiff mooring system that has no adjustment. In my opinion, this is an unfair comparison. The conclusions would have had more value if they were compared to a soft system that is not designed to contribute to optimize power production.

This issue was addressed in the previous response. The current paper presents a new innovative way for floating wind farm layout design and optimization. Therefore, we needed a benchmark that is accepted and currently used by both the research community and industry. Our benchmark for comparison needs to meet the current state-of-the-art design requirements; otherwise, the credibility of our results is highly compromised. We have updated the text: "The baseline mooring system design follows the current state-of-the-art mooring system design recommendations, and hence, it is valid to use it as a benchmark in this study."